# Evaporation suppression and energy balance of water reservoirs covered with self-assembling floating elements

Milad Aminzadeh[1,2], Peter Lehmann[1], Dani Or[1]

[1]Department of Environmental Systems Science, ETH Zurich, Switzerland

[2] now at Department of Civil Engineering, Isfahan University of Technology, Isfahan, Iran

*Correspondence to*: Milad Aminzadeh (m.aminzadeh@cc.iut.ac.ir)

**Abstract.** The growing pressure on natural fresh water resources and the projected climate variability are expected to increase the need for water storage during rainy periods. Evaporative losses present a challenge to the efficiency of water storage in reservoirs, especially in arid regions with chronic water shortages. Among the available methods for suppressing evaporative losses, self-assembling floating elements offer a simple and scalable solution especially for small reservoirs. The use of floating elements has often been empirically-based; we thus seek a framework for systematic consideration of floating element properties, local climate and reservoir conditions to better predict evaporative loss, energy balance and heat fluxes from covered water reservoirs. We linked the energy balance of the water column with energy considerations of the floating elements. Results suggest significant suppression of evaporative losses from covered reservoirs in which incoming radiative energy is partitioned to sensible and long wave fluxes that reduce latent heat flux and thus increase the Bowen ratio over covered water reservoirs. Model findings were consistent with laboratory scale observations using an uncovered and covered small basin. The study offers a physically-based framework for testing design scenarios in terms of evaporation suppression efficiency for various climatic conditions; hence strengthens the science in the basis of this important water resource conservation strategy.

# 1 Introduction

The competition over dwindling fresh water resources is expected to intensify with projected increase in human population and expansion of irrigated land (Assouline et al., 2015), and with changes in precipitation and drought patterns (Dai et al., 2011). Present global storage capacity for reservoirs > 0.1 km$^3$ is about 6200 km$^3$, with estimated total storage volume of 8070 km$^3$ when smaller reservoirs are considered, resulting in total evaporating surface area exceeding 300000 km$^2$ (Lehner et al., 2011). The reliance on water storage in reservoirs (Figure 1) is likely to increase to mitigate seasonal shortages due to projected precipitation variability, and to meet water needs for increased population and food production. By some estimates up to half of stored water in small reservoirs is lost to evaporation (Craig, 2005; Rost et al., 2008) thereby exacerbating the water scarcity problem. Interest in methods for suppressing evaporation has led to upsurge in the use of self-assembling floating covers over water reservoirs (e.g., Los Angeles reservoir in Sylmar, California); yet the selection, performance and implementation of such measures remain largely empirical. Recent studies (Assouline et al., 2011; Ruskowitz et al., 2014) have shown that evaporation suppression is a highly nonlinear process that depends on the properties of the covers (size, shape, radiative and thermal properties).

This study aims to provide a scientific basis for using self-assembling floating covers to suppress evaporative losses from reservoirs. The available strategies include deepening the water reservoirs (to reduce evaporative surface per stored volume), covering the surface, underground storage, or introducing wind breakers to reduce exchange with wind. Among these different measures for evaporation suppression, the use of self-assembling floating elements appears most promising for small-scale reservoirs due to its simplicity, cost-effectiveness and scalability (Craig, 2005; Assouline et al., 2011; Gallego-Elvira et al., 2012; Chaudhari and Chaudhari, 2015). Floating covers spontaneously rearrange in response to changes in water level or external conditions, e.g., wind (in contrast with chemical films that may break up due to the wave action, UV radiation, or biological activity).

Laboratory studies of evaporation from partially covered water surfaces (Assouline et al., 2010; Assouline et al., 2011) suggest a nonlinear relationship between the covered area fraction and

evaporative losses (see Figure 1 in Assouline et al. (2011)). These nonlinearities are attributed to vapor diffusion from water gaps across air viscous boundary layer (Schlünder, 1988; Shahraeeni et al., 2012; Haghighi et al., 2013) and potential feedback on the gap temperature (Aminzadeh and Or, 2013). The combined effects of gap size, spacing and thickness of the air boundary layer (Shahraeeni et al., 2012) support laboratory experimental results of Assouline et al. (2011) that have shown higher evaporation rates from small water gaps (per unit gap area) relative to evaporation rates from larger gaps (with similar uncovered surface fraction). These nonlinear relationships and additional energetic constraints must be considered in design and deployment of evaporation suppression floating covers.

The quantification of energy partitioning over partially covered water surfaces remains largely empirical with limited predictive capabilities beyond calibrated scenarios (Cooley, 1970; Assouline et al., 2011; Yao et al., 2010; Gallego-Elvira et al., 2011). Incoming radiative energy is intercepted primarily by the floating covers in which energy is mediated by cover geometry, radiative properties (albedo and emissivity), heat conduction and heat capacity of the material. The absorbed heat may be transferred to the water body in contact with floating covers, or return to the atmosphere as emitted long wave radiation and sensible heat flux. Interactions of floating elements with air flow regimes over the surface (turbulent or laminar) may generate complex aerodynamic patterns that affect sensible heat flux from surface elements.

The thermal coupling between floating cover elements and the water body has seldom been considered systematically by investigating surface energy balances for water and covers (Cooley, 1970). A few studies have considered this aspect via changes in heat storage of the water body as deduced from measurements (Gallego-Elvira et al., 2012). As the covered area fraction increases, the increase in intercepted radiative energy over the floating elements and their potential warming up may increase (lateral) heat fluxes towards water gaps thereby contributing to enhanced vapor flux from the uncovered water surface fraction (Aminzadeh and Or, 2017). Additionally, the decrease in the radiative energy penetrating into the water body affects the heat storage and aspects of biological activity within the reservoir. Hence, consideration of the energy balance over water surfaces covered by floating elements

is a critical ingredient for any design and management of evaporation suppression from water reservoirs that will be analyzed in this study.

The objectives of this study are to: (1) mechanistically model energy storage and surface fluxes of uncovered and partially covered water reservoirs; (2) consider the effect of cover properties on surface heat fluxes and radiative energy storage in a reservoir; and (3) predict evaporation suppression efficiency of floating covers.

In the following, theoretical considerations of energy balance for uncovered and partially covered water reservoirs are presented. We then investigate evaporation suppression efficiency of floating discs and their effects on surface heat fluxes and radiative energy storage.

## 2 Modeling framework

### 2.1 Energy balance and evaporation from uncovered water reservoirs

Before considering evaporation suppression from covered reservoirs, we first quantify fluxes from the uncovered reservoir as the reference state. The quantification of the temperature profile within the water body is the key to define surface heat fluxes and thus radiative energy storage into the reservoir. For simplicity, we employed a one-dimensional energy balance equation with subsurface radiation absorption and diffusive heat transfer including molecular and eddy thermal diffusivity to describe the vertical temperature profile in a reservoir according to (Dake and Harleman, 1969; Vercauteren et al., 2011):

$$\frac{\partial T_w}{\partial t} = \frac{\partial}{\partial z}\left(\left(\alpha_{T,w} + D_w\right)\frac{\partial T_w}{\partial z}\right) + \frac{Q(z,t)}{\rho_w c_w} \tag{1}$$

where $T_w$ is water temperature at depth $z$, $\alpha_{T,w}$ is molecular thermal diffusion, $D_w$ is eddy thermal diffusivity, $\rho_w$ and $c_w$ are density and specific heat of water, respectively. The heat source $Q$ accounts for the absorption of radiative flux within the water body and is a function of depth (light attenuation) and time (diurnal/seasonal variation of incoming radiation) (Dake and Harleman, 1969):

$$Q(z,t) = \eta(1-\beta)(1-\alpha_w)R_s(t)\,e^{-\eta z} \tag{2}$$

where $\beta$ is the absorption coefficient of incoming solar radiation ($R_s$) at the water surface, $\alpha_w$ is water surface albedo and $\eta$ is the light attenuation coefficient that is affected by the total suspended solids, dissolved organic matter and chlorophyll (Lee and Rast, 1997). For example, $\eta$ increases with increasing water turbidity. Alternatively, the heat source term can be quantified based on the dependence of light attenuation on wavelength (Rabl and Nielsen, 1975; Vercauteren et al., 2011). Among different formulations for eddy thermal diffusivity that governs heat transfer within the water body (McCormick and Scavia, 1981; Malacic, 1991; Vlasov and Kelley, 2014), we opt for the analytical representation based on Henderson-Seller (1985) which describes $D_w$ as a function of depth, density and friction velocity at the surface (that is a function of wind speed over the reservoir):

$$D_w = \frac{ku_s^* z}{P_0}\exp(-k^* z)[1+37R_i^2]^{-1} \tag{3}$$

where $k$ is von Karman's constant, $P_0$ is the neutral value of turbulent Prandtl number, $u_s^*$ is the friction velocity at the water surface that is characterized based on friction velocity of the air flow at the surface ($u_a^*$):

$$u_s^* = \sqrt{\frac{\rho_a}{\rho_w}}u_a^* \tag{4}$$

with air density $\rho_a$. The parameter $k^*$ is a function of latitude ($\phi$) and wind speed ($U$) (Henderson-Seller, 1985):

$$k^* = 6.6\sqrt{\sin\phi}\,U^{-1.84} \tag{5}$$

and $R_i$ is the Richardson number defined as (Henderson-Seller, 1985):

$$R_i = \frac{-1+\left(1+40N^2k^2z^2/(u_s^{*2}\exp(-2k^* z))\right)^{1/2}}{20} \tag{6a}$$

with buoyancy frequency $N$ :

$$N^2 = \frac{-g}{\rho_w}(\frac{\partial \rho_w}{\partial z}) \tag{6b}$$

The bottom boundary condition in deep reservoirs is often considered as a constant temperature or zero heat flux. In shallow reservoirs, one must consider the energy balance at the reservoir bottom and heat exchange with soil profile beneath. Hence, in a shallow reservoir with depth $D$, the energy balance at the bottom and related heat flux are expressed as:

$$\rho_w c_w (\alpha_{T,w} + D_w) \frac{\partial T_w}{\partial z}\bigg|_{z=D} = (1-\beta)(1-\alpha_w)R_{s,D}(t) + \frac{k_s}{Z}(T_{sZ} - T_D) \tag{7}$$

where $k_s$ is the effective thermal conductivity of the soil layer beneath the reservoir, $R_{s,D}$ is the shortwave radiation intercepted at the bottom of reservoir, $T_D$ is the bottom temperature of the reservoir (assumed similar to the water temperature at $z = D$ (Incropera and DeWitt, 2001)), and $T_{sZ}$ is a linearized soil temperature at thermal decay depth $Z$ (Shahraeeni and Or, 2011; Aminzadeh and Or, 2014). The water surface energy exchange expressed in terms of radiative, sensible and latent heat fluxes governs the surface boundary condition for Eq. (1):

$$\rho_w c_w \alpha_{T,w} \frac{\partial T_w}{\partial z}\bigg|_{z=0} = \beta(1-\alpha_w)R_s(t) + \sigma(\varepsilon_a T_a^4 - \varepsilon_w T_{ws}^4) + h_a(T_a - T_{ws}) - \frac{D_a L}{\delta}(C_s(T_{ws}) - C_a) \tag{8}$$

where $\alpha_w$ is water surface albedo, $\sigma$ is the Stefan-Boltzmann constant, $\varepsilon_a$ and $\varepsilon_w$ are atmospheric and water surface emissivity, respectively, $T_{ws}$ is water surface temperature, $T_a$ is the air temperature, $h_a$ is the sensible heat flux coefficient (see below), $D_a$ is the vapor diffusion coefficient in air, $L$ is the latent heat of vaporization, $\delta$ is the thickness of air boundary layer that is a function of wind speed (Haghighi and Or, 2013), $C_s$ is saturated vapor concentration at the water surface and $C_a$ is the vapor concentration in air mass above the boundary layer. The dependency of saturated vapor concentration on the water surface temperature (Eq. 8) through the Clausius-Clapeyron relation highlights the potential nonlinear evaporation enhancement with surface warming (Aminzadeh and Or, 2014). Note

that evaporative flux (driven by vapor concentration gradient) could alternatively be represented in terms of specific humidity. The sensible heat flux coefficient $h_a$ is quantified as (Gaikovich, 2000; Aminzadeh and Or, 2014; Haghighi and Or, 2015a):

$$h_a = \frac{k_a}{\delta} \tag{9}$$

in which $k_a$ is the air thermal conductivity.

Often, an unstable temperature profile develops in the water column where a cold water layer may form above a warmer layer due to subsurface radiation absorption; such conditions trigger convective mixing in natural reservoirs. Typically, mixing processes in the water body may require complex and higher dimensional modeling of flows; however, for simplicity, we opted for the 1D mixing approach of Dake

and Harleman (1969) that results in a uniform temperature within a mixed layer of water while conserving energy (see Figure 2):

$$\int_0^{h_m} (T_w - T_m)\, dz = 0 \tag{10}$$

where $T_m$ and $h_m$ are temperature and vertical thickness of the surface mixed layer, respectively. The solution of Eq. (1) results in vertical temperature profile, an important ingredient for quantifying surface

heat fluxes including evaporative loss from the reservoir (and for updating the mixed layer temperature).

The inflows and outflows of water in a reservoir may alter the heat storage of the water body, especially in multiuse reservoirs (e.g., water release for electrical energy production in dams). The net advected heat into the reservoir is characterized by the volume-weighted heat content of water inflows and

outflows (Moreo and Swancar, 2013):

$$Q_V = \sum_i \rho_w c_w V_i (T_i - \bar{T}) - \sum_e \rho_w c_w V_e (T_e - \bar{T}) \tag{11}$$

where $Q_V$ is the rate of change in heat content due to the changes in water budget of the reservoir; $V_i$ and $T_i$, and $V_e$ and $T_e$ are the rates and mean temperatures of inflows and outflows, respectively, and

$\overline{T}$ is the mean temperature of the reservoir. The parameter $Q_V$ can be considered in terms of a heat source/sink (e.g., similar to the radiation absorption) to investigate the effect of heat advection due to water exchanges on the energy balance and thus temperature profile in a reservoir.

We have neglected lateral conductive heat transfer in the reservoir assuming that the side area of the reservoir is small relative to its surface area (reflecting conditions in many shallow reservoirs where surface fluxes and subsurface radiation absorption dominate). This simplifying assumption enables focus on a simple 1D model for quantification of vertical temperature profile and thus surface heat fluxes from uncovered and covered shallow water bodies.

## 2.2 The energy balance of partially covered reservoirs

The use of floating cover elements aimed to suppress evaporative losses also modifies interactions of the reservoir surface with overlying air flow and thus wind-driven subsurface mixing patterns. The interception of radiative flux by the cover surface decreases radiation penetration into the water body shifting the energy partitioning to the cover surface. To simplify the analyses, we consider the energy balance of a reservoir covered by floating Styrofoam discs (similar to the laboratory experiments described in section 3.2). A covered reservoir surface (Figure 3a and b) is represented by a unit cell comprised of a floating disc surrounded by water gaps whereby the ratio of cover area to the total unit cell area defines the surface coverage (Figure 3c). We thus modify the surface boundary condition of the reservoir while retaining a simple 1D formulation and considering energy exchanges with the airflow and floating elements:

$$\rho_w c_w \alpha_{T,w} \left. \frac{\partial T_w}{\partial z} \right|_{z=0} = f_w \left( \beta(1-\alpha_w)R_s(t) + \sigma(\varepsilon_a T_a^4 - \varepsilon_w T_{ws}^4) + h_a(T_a - T_{ws}) - \varphi \frac{D_a L}{\delta_e}(C_s(T_{ws}) - C_a) \right) + f_c q_c \quad (12)$$

where $f_w$ and $f_c$ are the areal fractions of free and covered surface, respectively ($f_w = 1 - f_c$), and $\delta_e$ represents an effective air boundary layer thickness over the partially covered reservoir. The parameter $\varphi$ accounts for the aerodynamic enhancement of vapor flux from relatively small water gaps in comparison with the thickness of viscous sublayer (Assouline et al., 2011). Hence, the reduction of

vapor diffusion resistance from individual gaps (governed by the combined effect of gap size $a_g$, boundary layer thickness and lateral spacing) would enhance vapor diffusion and result in values of $\varphi$ $\geq 1$ that is defined as (Schlünder, 1988; Haghighi et al. 2013):

$$\varphi = \frac{1}{f_w + \dfrac{a_g}{\delta_e}\sqrt{\dfrac{f_w}{\pi}}\left(\sqrt{\dfrac{\pi}{4f_w}} - 1\right)} \tag{13}$$

Note that in this expression it was assumed that $a_g$ has a circular shape. This expression becomes effective for gap sizes much smaller than the boundary layer thickness.

Due to the strong lateral mixing induced by air flow at the reservoir surface (relative to the scale of water gaps), we assume a uniform horizontal temperature at the water surface that is defined based on the heat exchanges with air and conductive flux between floating elements and water surface ($q_c$) via

Eq. (12). Hence, the energy balance equation of the floating disc in the unit cell is used to quantify temperature distribution of the cover and thus the conductive heat exchange with water:

$$\frac{1}{\alpha_{T,c}}\frac{\partial T_c}{\partial t} = \frac{1}{r}\frac{\partial}{\partial r}(r\frac{\partial T_c}{\partial r}) + \frac{\partial^2 T_c}{\partial h^2} \tag{14}$$

in which $T_c$ is cover temperature at radial coordinate $r$ and thickness $h$, and $\alpha_{T,c}$ is molecular thermal diffusivity of cover material. The boundary condition at the surface and periphery of disc in contact

with air flow is governed by radiative and sensible heat fluxes. For the bottom of the disc in contact with water surface we assume that the temperature equals to water temperature. Simultaneous solution of Eqs. (1) and (14) with associated boundary condition (Eq. 12) enables quantification of temperature profiles in water body and floating elements that are linked via conductive heat flux ($q_c$).

The air flow friction velocity ($u_a^*$) and the effective thickness of viscous sublayer over the partially

covered reservoir ($\delta_e$) are determined using the analyses of Haghighi and Or (2015b) for evaporating

porous surfaces covered with bluff body obstacles obtained based on the theory of drag partitioning over rough surfaces (Shao and Yang, 2008; Nepf, 2012) (see appendix A for details).

In summary, the effect of floating elements on the energy balance of the reservoir and thus surface fluxes is seen by considering: 1) the energy balance of the water column which now receives less radiative energy in the presence of covers, 2) the energy balance of the cover and its thermal exchanges with water column, and 3) the heat and mass exchanges at the surface of unit cell (comprised of floating cover and water gap) with overlying air flow.

## 3 Materials and methods

### 3.1 Model evaluation for the uncovered reservoir

The energy balance Eqs. (1) and (14) were solved numerically using the finite difference method (forward time-central space scheme). The modeling results of vertical temperature profile and surface heat fluxes for the uncovered water reservoir were assessed primarily by using water temperatures and surface fluxes (radiative, evaporative and sensible heat fluxes) measured at Lake Mead, USA (Moreo and Swancar, 2013). The model evaluation for the uncovered water body serves as a "reference state" for studying effects of partial cover on heat storage and energy balance of large water reservoirs. We have used hourly meteorological data from Lake Mead (air temperature and humidity, wind speed, and solar radiation) obtained from March 2010 to February 2011 to reproduce the evolution of water temperature profiles and associated heat fluxes. The thermal and radiative properties of the lake used in the model are listed in Table 1.

### 3.2 Laboratory experiments of evaporation suppression using floating discs

In the absence of "reservoir scale" data for model validation of a covered reservoir (e.g., potential data sets from Ivanhoe Los Angeles reservoir are not yet publically available), we designed a series of experimental studies of evaporation suppression from a small water basin covered with floating discs at laboratory scale (Figure 4). The main purpose was to systematically vary external forcing (wind, radiation and combination) towards gaining new insights into energy partitioning at the surface of

covered water bodies (the full scope of the laboratory study will be reported elsewhere). A subset of these laboratory experiments was used to provide a preliminary evaluation of the model for covered surface to improve understanding of reservoir scale modeling results.

A square-shaped water reservoir of 1.44 m$^2$ area and 0.16 m depth (mounted on a balance to measure mass loss) was covered with Styrofoam discs of 0.2 m diameter and 0.02 m thickness. The black or white colored discs covered 91% of the water surface. Wind velocities controlled using an upstream wind tunnel and shortwave radiation by four Xenon light sources were used independently and in combination to create different evaporative forcing (i.e., wind, radiation and wind+radiation). The resulting evaporation rates were determined by measuring the mass of the water basin. The air temperature, relative humidity and wind velocity were also monitored above the covered surfaces. An infrared camera (FLIR SC6000, USA) recorded the surface temperature of the covered reservoir with a spatial resolution of 0.8 mm. We conducted a series of experiments in which external boundary conditions (forcing) such as constant wind (~2.3 m/s) without radiation, radiation (~350 W/m$^2$) without wind, and combination of radiation and wind were maintained for two days to permit equilibration and convergence to steady state conditions. Similar series of conditions were applied to basin covered with white or black floating discs and to the same uncovered basin.

### 3.3 Modeling the energy balance of a partially covered reservoir

The model was used to evaluate a hypothetical covered reservoir using meteorological variables obtained from European Fluxes Database Cluster (http://www.europe-fluxdata.eu/home), with covers that resemble those used in the laboratory experiments. We have used half-hourly meteorological data including air temperature, relative humidity, wind speed and radiation for Majadas (Spain) representing conditions in a dry region with significant atmospheric evaporative demand for the water year from March 1, 2004 to March 1, 2005. The model was used to study potential effects of floating disc-shaped elements (diameter of 0.2 m and thickness of 0.02 m) on heat fluxes and water temperature profiles within the hypothetical reservoir with a depth of 10 m. The vertical simulation domain was comprised of 626 equally-spaced grid points at 0.016 m spacing. Initially, the reservoir was assumed to have a

uniform vertical temperature of 11 $^\circ$C with zero heat exchange at the bottom boundary. Details of the floating cover thermal and radiative properties are presented in Table 1.

## 4 Results

### 4.1 Energy budget and evaporation from uncovered water reservoirs – model application

We first assessed the modeling results of water temperatures and surface fluxes for the uncovered reservoir considering the Lake Mead data. Model estimates of mean monthly temperature profiles were compared with measured water temperature profiles in Lake Mead as depicted in Figure 5. The comparison illustrates that the model was able to capture the temperature dynamics in the lake reasonably well (with slight underestimation in late summer). The measured and simulated fluxes are summarized in Table 2 showing relative errors of 27% and 13% between modeled and measured annual values for net radiation ($R_n$) and evaporation ($LE$) fluxes, respectively. Given the simplifying assumptions, the model overestimated the sensible heat flux ($H$) reported by Moreo and Swancar (2013) based on the Bowen ratio method with associated energy closure considerations (Foken, 2008; Kalma et al., 2008).

### 4.2 Laboratory evaporation suppression experiments

The evaporation rates from the laboratory basin were obtained directly from a digital balance, whereas the surface temperature dynamics were recorded by IR thermography as depicted in Figures 6 and 7, respectively. Surprisingly, the evaporation measurements in Figure 6a show that irrespective of the type of external forcing (wind and radiation) or the color of the floating discs, the resulting evaporation rate from covered water surfaces was about 20% of the uncovered surface. The corresponding evaporation suppression efficiency of 80% is less than the cover fraction of 91%. This reduced efficiency is attributed partially to the increased surface temperature of the water between the discs compared to the uncovered water reservoir (shown in Figure 7). Subsequently we have used the laboratory forcing (wind, radiation, air temperature and humidity) in the model to describe the evaporative losses and capture temperature dynamics over the surface of uncovered and covered water basin. Despite the small

size of the basin (and scale mismatch with reservoir scale model), the simulations were in good agreement with evaporative mass loss rates (Figure 6b) and IR surface measurements (Figure 7). This limited experimental evidence highlighted the potential applicability of the model for quantifying evaporation suppression and predicting dynamics of surface temperature that is in the core of energy partitioning over covered water bodies.

The theoretical results supported by laboratory measurements clearly demonstrated that the main effect of floating covers on evaporation suppression and energy partitioning was concentrated at the surface and thus upheld the focus on the top boundary condition for the full reservoir scale model reported in this study. Nevertheless, we note that these experiments may not reflect influences of temperature profiles, heat storage, mixing and ground flux that could potentially affect water temperature and, in turn, the top boundary of the reservoir.

### 4.3 Evaporation and energy budget of partially covered water reservoirs

Model predictions for the evolution of mean daily temperature profile of uncovered and covered reservoir using white and black Styrofoam discs in a (hypothetical) reservoir with depth of 10 m are depicted in Figure 8. The stable temperature profile and slow heat uptake during spring results in a gradual temperature increase especially in uncovered reservoir. As expected, the highest water temperature of uncovered reservoir occurs during summer with a warm layer of water at top of the reservoir whose temperature decreases monotonically to the bottom. On the other hand, the onset of convective thermal mixings in fall and low radiative flux rapidly yield an almost uniform temperature profile in winter and beginning of spring. The reservoir was then assumed to be covered by Styrofoam discs with diameter of 0.2 m and thickness of 0.02 m providing a surface coverage of 0.91 (maximum packing of discs). Due to the geometry of floating elements and their density on the surface (cover areal fraction), the effective thickness of air boundary layer over the covered surface was calculated similar to the thickness of boundary layer over uncovered water reservoir (appendix A).

The mean daily temperature profiles of the reservoir covered with white and black discs depicted in Figures 8 clearly demonstrate that covering the surface with floating elements yields a much colder

reservoir. Surprisingly, despite large difference in surface albedo of black and white discs (see Table 1) and thus different cover surface temperatures (Figure 9), the resulting water temperature profile did not vary much between reservoirs covered with these two types of floating discs.

In the following, we investigate the effect of surface coverage and cover properties on the evolution of surface heat fluxes.

### 4.3.1 Energy partitioning and surface fluxes from partially covered reservoirs

The evolution of surface heat fluxes over the uncovered and partially covered reservoir is shown in Figure 10. The reflection of incoming shortwave radiation by the covers resulted in a decrease in net radiative flux of the covered reservoir. The impact of surface albedo on net radiative flux is evident with changing the cover color, yielding lower net radiation over the reservoir covered with white discs relative to the reservoir that is covered with black discs.

The effect of floating discs on evaporation from the reservoir is illustrated in Figure 10c. It shows that discs significantly suppress evaporative flux relative to uncovered water reservoir especially during summer. The substantial decrease in evaporative flux from covered reservoir with concurrent increase in sensible heat flux (due to the high cover temperature) results in a higher Bowen ratio over the covered reservoir relative to water surfaces (Priestley and Taylor, 1972). Interestingly, the color of the floating discs did not affect evaporation suppression from the covered reservoir; hence, the higher sensible heat flux from the black discs yields higher Bowen ratio relative to the white discs scenario. A summary of mean annual surface heat fluxes for uncovered and covered reservoir is presented in Table 3.

The ratio of heat storage in the water body relative to the net radiation over the surface of the uncovered and partially covered reservoir is shown in Figure 11. To compute the heat storage we have chosen the initial (assumed uniform) temperature profile at the beginning of the water year (March 1) as a reference. Such a reference state is motivated by measurements in Lake Mead (Figure 5). The heat storage is then calculated by integrating changes in the temperature profile relative to the reference (and

water heat capacity). At beginning of the year, the ratio of heat storage to net radiation is sensitive to temperature variations close to the surface showing large fluctuations. After an equilibration period, Figure 11 shows a maximum value of the ratio in the summer for the uncovered water reservoir before decreasing in the fall, following the annual variation of radiative flux. For the partially covered reservoir, the ratio remains nearly constant with only a slight increase during the summer. Moreover, the lower net radiative flux of the reservoir covered with white discs (Figure 10) results in higher values of the ratio of heat storage to the net radiation while subsurface heat storage does not change significantly with changing color of floating discs.

We have also investigated the effect of reservoir depth on energy storage and surface heat fluxes and a summary of results is provided in appendix B.

## 4.3.2 Evaporation suppression efficiency of floating covers

Self-assembling floating discs effectively cover the reservoir and decrease water surface exposure to the atmosphere. We plotted the ratios of evaporative fluxes from covered water reservoirs relative to uncovered water surface ($E_c / E$) to quantify evaporation suppression efficiency of the floating discs (i.e., $\varepsilon = 1 - E_c / E$). The results in Figure 12 demonstrate that application of discs yields more than 80% drop in evaporative loss from the reservoir ($E_c / E < 0.2$) with highest efficiency during summer. This result was obtained based on the 1D modeling of vapor flux ($\varphi = 1$) from relatively big water gaps between neighboring discs (diameter of 0.2 m) representing the upper bound of evaporation suppression efficiency. However, under certain conditions where the boundary layer thickness (often in the order of a few millimeters (Haghighi and Or, 2013)) is comparable with gap size ($a_g / \delta_e$ in Eq. 13), enhancement of vapor flux from individual gaps may decrease the suppression efficiency ($\varphi > 1$).

## 5 Discussion

### 5.1 The energy balance of covered reservoirs

The physically-based model describes various effects of floating element placement on the energy budget and surface fluxes of water reservoirs and the great potential for suppressing evaporative losses using such a simple method. Our modeling results demonstrated that covering the surface with modular floating elements yields a colder reservoir relative to the uncovered scenario due to the interception of incoming radiative flux by the cover surface shifting the energy partitioning to the reservoir surface. Despite significant difference in surface albedo of black and white discs, the water temperature profile of the covered reservoir was similar. We attribute this to the strong thermal insulation of the Styrofoam elements that effectively decoupled the top surface of the covers (that may attain different temperatures based on color) from the temperature and fluxes on the water surface.

In other words, the low thermal diffusivity of Styrofoam discs resulted in negligible heat conduction to the water body, whereas the intercepted radiative flux on the cover surfaces (especially the black) results in considerable increase of cover temperature (Figure 9a) with higher sensible heat and long wave radiate exchange. In contrast with expectation, the radiative properties of the floating covers did not affect the water surface temperature and the use of thermally insulating covers leaks only small amounts of heat to the water (Figure 9b) that mildly influences the evaporative flux from covered reservoir (Figure 10c). These results have been confirmed in laboratory experiments for the basin covered with white or black discs where the evaporative fluxes from the covered surface were 20% of the uncovered regardless of the cover color and forcing (see Figure 6a).

### 5.2 Evaporation suppression in covered reservoirs

The reduction of evaporating area on the surface of covered reservoirs primarily suppresses evaporative loss from the water body. Considering the 1D modeling of vapor flux in the present study, the decrease in evaporative loss is expected to be equal to the covered area fraction. However, the evaporation ratio larger than the uncovered areal fraction (0.09) in Figure 12 is attributed to the higher water surface

temperature in gaps between floating elements relative to the uncovered water surface as illustrated in Figure 9b. An interesting feedback mechanism may play a role in evaporation suppressing efficiency where high evaporative fluxes from uncovered water reservoirs may result in more surface temperature depression and thus lower saturated vapor concentration relative to the vapor concentration at the surface of water gaps over partially covered reservoirs. In addition, conductive heat fluxes from cover elements to the water surface could potentially contribute in increase of water surface temperature depending on thermal properties of the cover material. The higher gap temperature relative to the uncovered water obtained from the modeling was also observed in laboratory experiments (Figure 7a) supporting the nonlinearity of evaporation suppression from partially covered reservoirs.

Although we assumed that air temperature and humidity (obtained from European Fluxes Database for the numerical experiment) are the same over uncovered and partially covered reservoir, it is important to note that the higher sensible heat flux over covered reservoir could locally increase air temperature in contact with water gaps that, in turn, enhances evaporative loss from covered reservoir and decreases evaporation suppression efficiency. In addition to the physical considerations of the energy balance and evaporation suppression in covered reservoirs, further investigations including the ecological aspects and cost efficiency discussed below are needed to provide a comprehensive assessment for application of floating covers.

**5.3 Ecological considerations**

Reservoirs may serve multiple functions including the support of various ecosystems; hence, the introduction of opaque floating covers to supress evaporation may alter water temperature, light penetration and gas exchange all affecting the life in the reservoir. The full consideration of ecological targets is beyond the scope of this study, clearly, certain parameters could be included in the cover design and management to limit adverse impacts on the ecology of the water body (in some cases, a cover may supress toxic algal blooms in a reservoir). For example, here we consider effects of floating covers on gas exchange across the air-water interface as a function of uncovered fraction ($f_w$). The

oxygen transfer at the surface of reservoir ($F_{s,O_2}$) is expressed as (Stefan et al., 1995; Schladow et al., 2002):

$$F_{s,O_2} = f_w\, k_{s,O_2}(C_{e,O_2} - C_{w,O_2})$$ (15)

where $k_{s,O_2}$ is the oxygen transfer coefficient, and $C_{e,O_2}$ and $C_{w,O_2}$ are equilibrium oxygen concentration and oxygen concentration in the surface layer, respectively. The dissolved oxygen in the water body is consumed by aerobic organisms (e.g., fish and aquatic microorganisms) and affects various chemical reactions in a reservoir (Stefan et al., 1995). The mechanical sheltering impact of surface covers that dampens wind-driven mixing at the surface may affect air-water oxygen exchange and transport in water column yielding a stratified oxygen profile in the reservoir. Although the interception of radiative flux by the cover surface decreases subsurface radiation absorption and results in a colder reservoir that may enhance oxygen solubility in water (Wilkinson et al., 2015), the reduction of radiation absorption limits convective mixing driven by unstable temperature profiles and intensifies a stratified oxygen distribution. Moreover, the photosynthesis by aquatic plants and microorganisms in darker and colder reservoirs covered with floating elements decreases which then affects the oxygen budget according to the oxygen transfer equation (Stefan et al., 1995):

$$\frac{\partial C}{\partial t} = \frac{\partial}{\partial z}\left( \left( \alpha_{O_2} + D_w \right) \frac{\partial C}{\partial z} \right) + P_{O_2} - R_{O_2}$$ (16)

where $C$ is oxygen concentration at time $t$ and depth $z$, $\alpha_{O_2}$ is molecular oxygen diffusion, and $P_{O_2}$ and $R_{O_2}$ are oxygen production by photosynthesis and consumption due to biological activities within the water body, respectively. In summary, exchange rates and oxygen production and concentration in a reservoir are strongly dependent on water temperature, radiative flux, transport processes and nutrients that are likely to be influenced by surface coverage. Note that ecological considerations of covered reservoirs are not limited to aquatic organisms and additional aspects including accessibility of birds and wildlife should also be investigated. Such ecological objectives become part of the reservoir cover

design and evaporation suppression must be balanced by ecological and also economic constraints as discussed next.

## 5.4 Costs and water saving

The significant reduction in evaporative loss from the reservoir could be gauged by direct economic impact in terms of cost of alternate source of water where available. The economic efficiency of evaporation suppression depends on the costs of construction ($P_c$ [\$/m$^2$]), annual maintenance of covers ($P_m$ [\$/m$^2$year]), alternate water cost ($w$ [\$/m$^3$]), annual evaporation from the uncovered reservoir surface ($E$ [m/year]), and evaporation suppression efficiency of floating covers ($\varepsilon$) (Cooley, 1983; Assouline et al., 2011). Assuming a life span of $Y$ years for the floating elements, the economic efficiency per unit area of reservoir ($\zeta$ [\$/m$^2$]) is estimated as:

$$\zeta = Y(\varepsilon wE - P_m) - P_c \tag{17}$$

We thus calculate $\zeta$ for the hypothetical reservoir presented in section 3.3 with annual evaporative losses for uncovered surface $E$ =1.6 m/year (see Table 3), and estimated cover efficiency $\varepsilon$ =0.8. Considering water price $w$ =1 \$/m$^3$ (e.g., seawater reverse osmosis costs are in the range of 0.5 to 3 \$/m$^3$ (Gilau and Small, 2008; Guler et al., 2015)) floating cover cost $P_c$ =5 \$/m$^2$ (based on commercially available HDPE floating balls) and cover maintenance cost $P_m$ =0.1 \$/m$^2$year, the economic efficiency of such floating elements for a period of 5 years is ~1 \$/m$^2$. Hence, for a reservoir with 100×100 m$^2$ surface area, water costs saving equivalent to \$10000 is feasible in 5 years operation (along with 64000 m$^3$ of water protected from evaporation).

Tacit in this standard estimate is availability of an alternate water source (e.g., desalinated water); whereas in many regions in the world with poor infrastructure and acute water shortages, the value of evaporation suppression may transcend such estimates and the real measure could be expressed in terms of livestock supported by the additional water or avoidance of crop failure. Water scarcity and droughts may exacerbate water shortage problems and transboundary (or regional) conflicts over shared water

resources. Some of these political challenges could be alleviated by promotion of efficient local storage using cost effective evaporation loss mitigation measures (such as floating covers).

## 5.5 Improvement of the modeling approach

Many aspects of the hydrodynamics and turbulent conditions associated with atmospheric stability over the evaporating reservoir surface were not explicitly addressed in the present study. In a recent study of soil surface evaporation, Haghighi and Or (2015c) have linked effects of different stability conditions in the Monin-Obukhov similarity (MOST) atmospheric turbulent profiles with the surface boundary layer approach used in the present work. The study offered corrections for adjusting the viscous sublayer and thus the effects of atmospheric stability conditions on heat and vapor exchange with surfaces. Elements of the analyses of Haghighi and Or (2015c) could be incorporated into the modeling of surface-atmosphere exchanges over partially covered water reservoirs. Such an analysis would be warranted once we resolve important aspects of the effects of floating elements on features of the viscous sublayer over the partially covered surface. At present, the effects of floating element shapes and cover density on surface shear stresses (in the air and in the water body), impacts of inflows-outflows, bottom topography and breaking waves have not been implemented and are expected to affect surface condition and subsurface turbulent mixing and thus modify effective eddy diffusivity. Moreover, the model parametrization should consider the depth dependency of eddy thermal diffusivity and the effect of reservoir depth on largest thermal eddies that could develop in the water body. The availability of data from covered ponds and larger reservoirs would provide the impetus to systematically address these important ingredients and improve the predictive framework for application of modular covers in controlling evaporative losses from water bodies.

As commented above, the simple 1D energy and mass flux model has tacitly neglected lateral conductive fluxes between the water body and sides of the reservoir which was deemed a reasonable approximation for shallow reservoirs and where floating elements dominate surface fluxes. Energy balance errors incurred due to lateral heat fluxes in small reservoirs (e.g., agricultural ponds) warrant

special studies (motivated by measurements) to improve energy closure and provide reliable estimates of surface fluxes and evaporation suppression efficiency.

## 6 Summary and conclusions

The harnessing of the great potential of using floating elements to suppress evaporative losses from water reservoirs requires a transition from anecdotal and empirical-based designs and applications into employing a systematic modeling framework capable of integrating local climatic variables with reservoir and cover properties in a predictive manner. To meet the design and prediction challenges, we developed and tested a simple energy balance model for quantifying surface fluxes and vertical temperature profiles in a water reservoir. The simultaneous solution of energy balance equations for the water body and floating elements linked heat exchanges between cover and water surface and enabled quantification of surface heat fluxes and energy storage within the water body. Due to the absence of data from covered surfaces at the reservoir scale, we combined experimental information from a laboratory scale water basin covered with different floating elements (white or black Styrofoam discs) and subjected to a range of different forcing for model testing. The consistency of model findings with the experimental evidence obtained under controlled laboratory conditions provided valuable insights towards better quantification of energy partitioning dynamics over covered water bodies. The modeling results for a hypothetical reservoir covered with similar floating covers (Styrofoam discs) provided an opportunity for evaluating (theoretically) the response of a realistic reservoir over a full water year. The results demonstrated that interception of radiative flux by floating covers significantly decreases subsurface radiative energy absorption in covered reservoirs yielding colder water temperatures relative to uncovered water reservoirs. The lower water temperatures and reduced radiative energy storage for covered reservoirs could alter dissolved oxygen in the water body and exchange rates with the atmosphere, hence affecting aquatic life. The intercepted radiative flux on the surface of floating elements that primarily increases cover temperature is released in form of sensible heat flux and long wave radiation into the atmosphere depending on the cover thermal and radiative properties. The increased sensible heat flux could raise local air temperature over water gaps and contribute in the

enhancement of evaporative losses from individual gaps. Such nuanced aspects of energy partitioning over covered surfaces not investigated in the present study may decrease suppression efficiency of floating elements.

The modeling results (supported by laboratory experiments in a shallow basin) suggest that evaporation from the covered reservoir was reduced by about 80% relative to the uncovered water surface. Interestingly, the model shows that floating covers with low thermal conductivity are energetically decoupled from the water surface. Consequently, changes in cover color (affecting albedo) did not significantly modify the evaporative flux (a result that was also observed in laboratory experiments). The main effect of cover color was expressed either in the increase in cover temperature with associated increase in sensible and longwave radiative fluxes for the black cover; or the increased reflectance of shortwave radiation for the white covers (and lower cover temperatures). The reduction in evaporative fluxes and the higher sensible heat flux over partially covered reservoir may result in a significantly higher Bowen ratio over the covered relative to uncovered water surfaces (Priestley and Taylor, 1972).

Floating elements efficiently suppress evaporative losses from water reservoirs while altering the energy storage within the reservoir and potentially reduce oxygen exchange at the water-air interface. Notwithstanding theoretical considerations of evaporation and energy balance of covered reservoirs in the present study that were primarily aimed at developing a physically-based framework for design purposes, ecological consequences of such evaporative loss mitigation strategy must consider effects of reduced light and lower oxygen exchange on biota, especially in multiuse reservoirs. The model provides a useful tool for investigating effects of partial coverage and reservoir depth on surface fluxes and specific energy storage in the water body, and thus may provide design and management guidelines for different objectives, ranging from evaporation suppression to other ecological goals. The study highlights the need for field scale experimental studies of evaporation and energy fluxes from partially covered reservoirs (different covers and climatic conditions) towards generalization of the results and development of new insights, and for critical evaluation of key assumptions.

**Acknowledgments**

Financial support by the Swiss National Science Foundation (200021-172493) is gratefully acknowledged. The authors are grateful for access to the data from European Fluxes Database Cluster (ES-LMa 2004 and 2005: CarboEuropeIP (EU-FP6)). We thank Martin Schmid (EAWAG) for constructive discussions, and greatly appreciate the technical assistance of Hans Wunderli, Daniel Breitenstein, Martina Sommer, and Hannah Wey in the laboratory experiments, and the insightful inputs of Ali Ebrahimi (MIT) in various modeling aspects.

**Appendix A: Effective boundary layer thickness over covered reservoirs**

We use the analysis of Haghighi and Or (2015b) based on the theory of drag partitioning over rough surfaces (Shao and Yang, 2008; Nepf, 2012) to obtain the effective thickness of viscous sublayer over the covered reservoir:

$$\delta_e = g(\alpha)\frac{v}{u_a^*} \tag{A1}$$

where $g(\alpha)$ describes the effect of eddy characteristics (=21 for a practical range), $v$ is the kinematic viscosity of air, and $u_a^*$ is the air flow friction velocity:

$$u_a^* = U\sqrt{f_r\lambda(1-f_c)C_{rg} + \left(f_s(1-f_c)+f_gf_c\right)C_{sg}} \tag{A2}$$

where $U$ is air flow velocity, and $\lambda$ is the frontal area index that is a function of disc diameter ($d$) and height ($H$):

$$\lambda = N\,d\,H \tag{A3}$$

with $N$ as the number of discs per unit area; $C_{rg} = \gamma C_{sg}$ and $C_{sg}$ are drag coefficients of discs and uncovered surface, respectively with $\lambda = 0$:

$$C_{sg} = \left( k / \ln \left( z_U / z_{0s} \right) \right)^2 \tag{A4}$$

where $z_U$ and $z_{0s}$ are reference height for measurement of wind velocity and roughness length of uncovered surface, respectively. The parameters $f_r$, $f_s$, and $f_g$ are defined as:

$$f_r = \exp \left( -\frac{a_r \lambda}{(1 - f_c)^m} \right) \tag{A5}$$

$$f_s = \exp \left( -\frac{a_s \lambda}{(1 - f_c)^m} \right) \tag{A6}$$

$$f_g = 1 + \left( \frac{C_{sgc}}{C_{sg}} - 1 \right) f_c \tag{A7}$$

with $a_s = 5$, $a_r = 3$, and $m = 0.1$. The drag coefficient on the surface of disc $C_{sgc}$ is expressed as:

$$C_{sgc} = \left( k / \ln \left( \frac{z_U - H}{z_{0s}} \right) \right)^2 \tag{A8}$$

By increasing $\lambda$ from zero (uncovered surface) to $\lambda \approx 0.2$, the interaction of overlying air flow with floating elements results in formation of smaller scale eddies which then decrease the effective thickness of viscous sublayer. Increasing $\lambda$ more than 0.2 reduces air flow penetration into the gaps which thus traps air between floating elements and forms relatively thick boundary layer in the order of element's height. Figure A1 depicts the effect of cover geometry (diameter and height) on the effective thickness of boundary layer.

**Appendix B: The effect of reservoir depth on energy balance**

We investigated the effect of reservoir depth on the energy balance and surface heat fluxes considering shallow (3 m) and deep (10 m) hypothetical reservoirs for the conditions in Majadas, Spain (March 2004 to March 2005). The bottom boundary condition was assumed to follow a linear heat flux to the underlining soil (Shahraeeni and Or, 2011). Although (as expected) the specific energy storage (storage

per volume of reservoir) was higher in the shallow reservoir, the surface temperature and heat fluxes were similar for the shallow and deep reservoirs (Table B1). Considering similar aerodynamic conditions at the surface, the similarity in surface fluxes of shallow and deep reservoirs indicate that surface temperatures were similar (e.g., uncovered water surface temperature depicted in Figure B1). These results highlight the dominance of atmospheric forcing in adjusting surface temperature and thus surface heat fluxes whereas the effect of reservoir depth is reflected in the specific energy storage and heat flux at the bottom (especially in uncovered reservoirs), $G$, which is governed by the bottom temperature of the reservoir (Figure B1).

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

**Table 1: The physical properties of water and the Styrofoam discs (white and black surfaces) used for modeling.**

|  | Specific heat (J/kg K) | Emissivity | Albedo | Thermal conductivity (W/m K) | Molecular thermal diffusivity (m²/s) |
|---|---|---|---|---|---|
| Water | 4200 | 0.95 | 0.05 | 0.6 | $1.43 \times 10^{-7}$ |
| Discs | 1130 | 0.85 | white: 0.6 black: 0.1 | 0.03 | $3.9 \times 10^{-8}$ |

**Table 2: Comparison of modeled and measured annual surface energy balance components for (uncovered) Lake Mead (2010-2011).**

|  | $R_n$ (W/m²) | LE (W/m²) | E (mm/day) | H (W/m²) |
|---|---|---|---|---|
| Measurements (Bowen ratio EB) | 147 | 170 | 5.95 | -18 |
| Model estimates | 187 | 148 | 5.2 | -36 |

**Table 3: Modeled annual surface heat fluxes of uncovered and covered hypothetical reservoir using meteorological data from European Fluxes Database for Majadas, Spain (March 2004 to March 2005).**

|  | $R_n$ (W/m²) | LE (W/m²) | E (mm/day) | E (mm/year) | H (W/m²) |
|---|---|---|---|---|---|
| Uncovered | 147.9 | 127.3 | 4.48 | 1635 | -65.1 |
| Covered with black discs | 122.1 | 14.5 | 0.51 | 187 | 94.9 |
| Covered with white discs | 54.8 | 12.9 | 0.45 | 167 | 30.7 |

**Table B1: The effect of reservoir depth on heat fluxes and specific storage of uncovered and covered reservoirs.**

| | | $R_n$ (W/m$^2$) | H (W/m$^2$) | G (W/m$^2$) | E (mm/day) | Storage: June-Sep (MJ/m$^3$) |
|---|---|---|---|---|---|---|
| 3 m depth | Uncovered | 148.7 | -67 | 29.2 | 4.43 | **25** |
| | Covered with black discs | 122.9 | 92.7 | 2.2 | 0.44 | 2.1 |
| | Covered with white discs | 55.4 | 28.5 | 1.8 | 0.39 | 1.6 |
| 10 m depth | Uncovered | 148.1 | -65.6 | 20.4 | 4.46 | **18.4** |
| | Covered with black discs | 122.5 | 93.8 | 2.2 | 0.47 | 1.9 |
| | Covered with white discs | 55.1 | 29.5 | 2.1 | 0.41 | 1.3 |

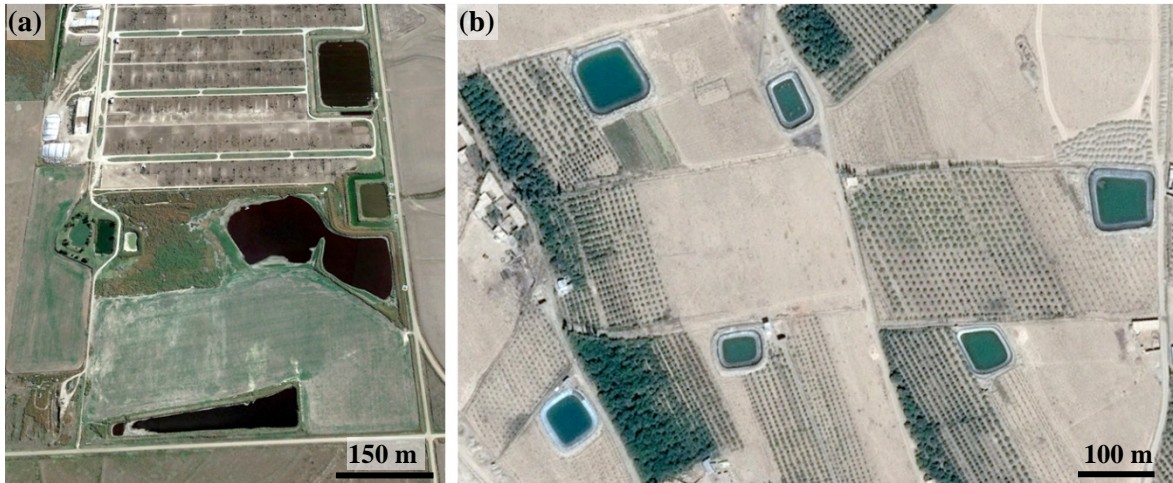

**Figure 1: The growing number of small reservoirs for local supply during dry periods highlights the need for evaporation suppression measures to conserve water (satellite images from (a) Hanston, Kansas, US, and (b) Shahrood, Iran; reproduced from Google Earth (2017)).**

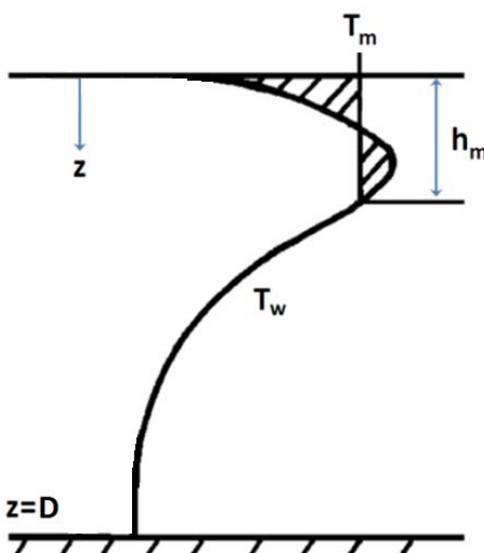

**Figure 2: Convective mixing at the surface of water reservoir of depth *D* due to the unstable temperature profile associated with subsurface radiation absorption (adapted from Dake and Harleman (1969)). Based on Eq. (10), the hatched areas on the left and right hand side of $T_m$ are equal representing transfer of subsurface heat accumulation to the surface.**

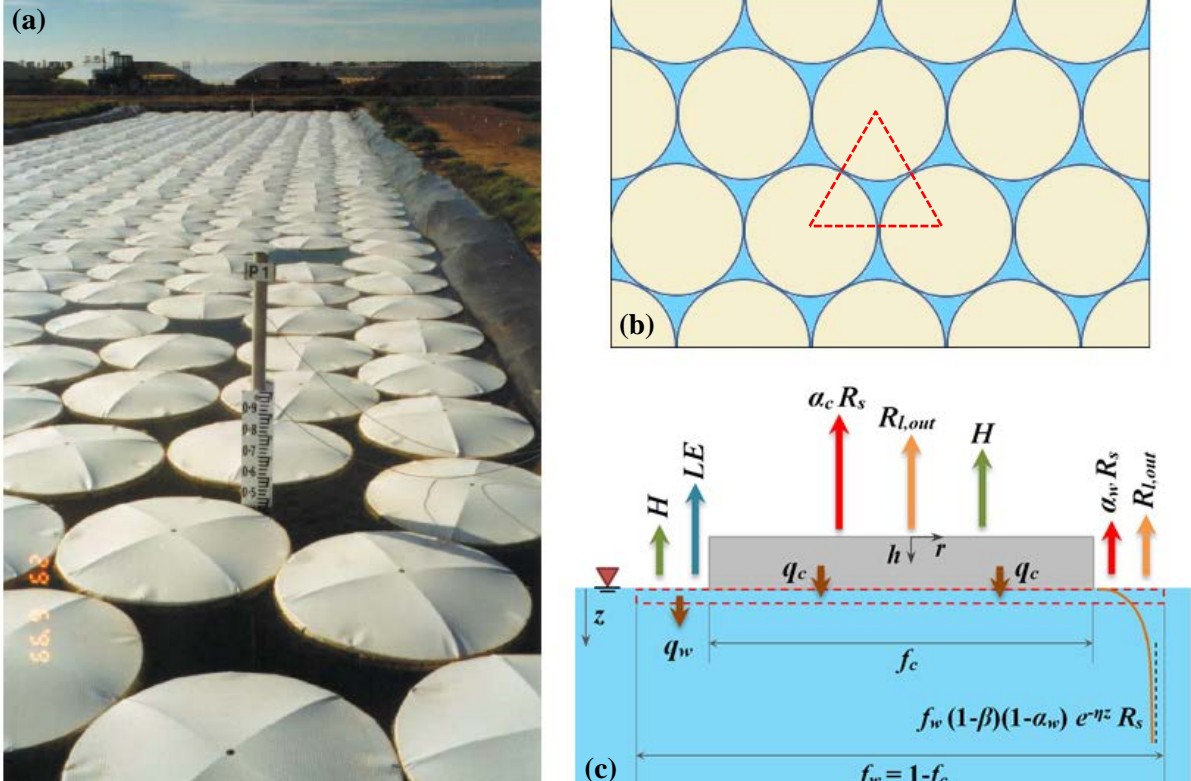

**Figure 3: (a) Application of floating discs in evaporation suppression from water reservoirs (adapted from Assouline et al. (2011)); (b) top view of reservoir surface covered with discs; due the geometrical constraints, dense packing of discs provides a surface coverage of 0.91 (inferred from the depicted triangle with side lengths equal to disc diameter); (c) schematic representation of subsurface radiation attenuation (the curve with associated expression) and surface heat fluxes in a representative unit cell including a floating element and free water surrounding it with areal fractions of $f_c$ and $f_w$, respectively (Eq. 12). See section 2 for definition of the various parameters.**

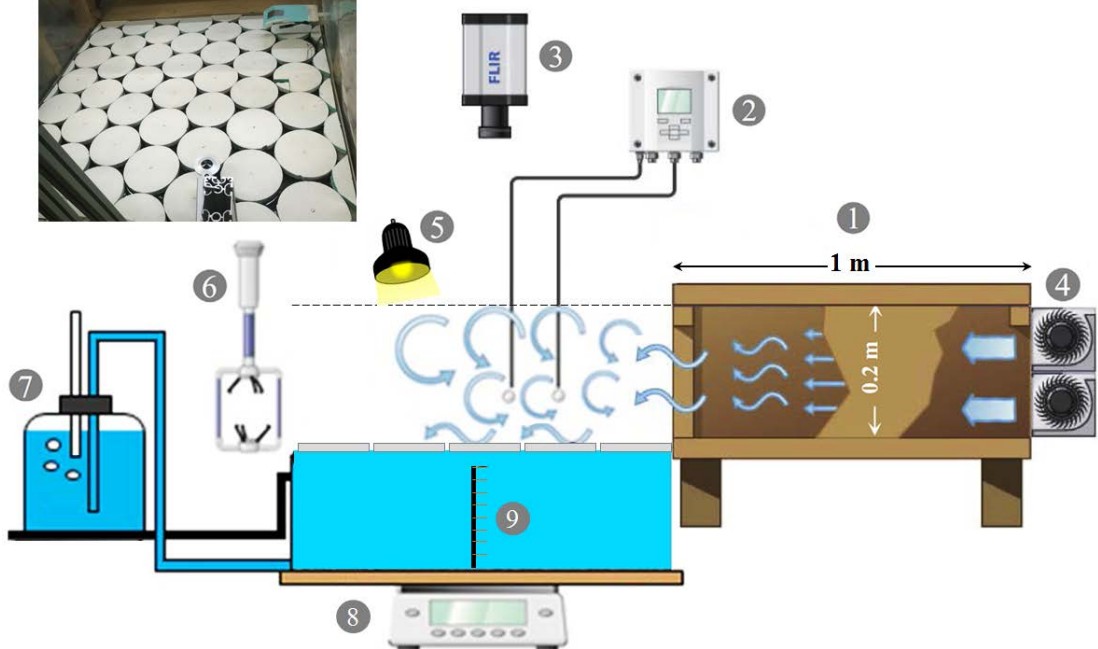

**Figure 4: Experimental setup for evaporation suppression measurements from a small water basin covered with floating discs: (1) wind tunnel, (2) air temperature and humidity sensors (Vaisala HUMICAP, HMT337, Finland), (3) IR camera (FLIR SC6000, USA), (4) tunable fans generating wind flow, (5) xenon lamps for shortwave radiation, (6) high-frequency 3D sonic anemometer (WindMaster, Gill Instruments Ltd., The Netherlands), (7) Mariotte bottle to adjust water level, (8) balance to determine mass loss and evaporation rates, (9) temperature sensors in water body.**

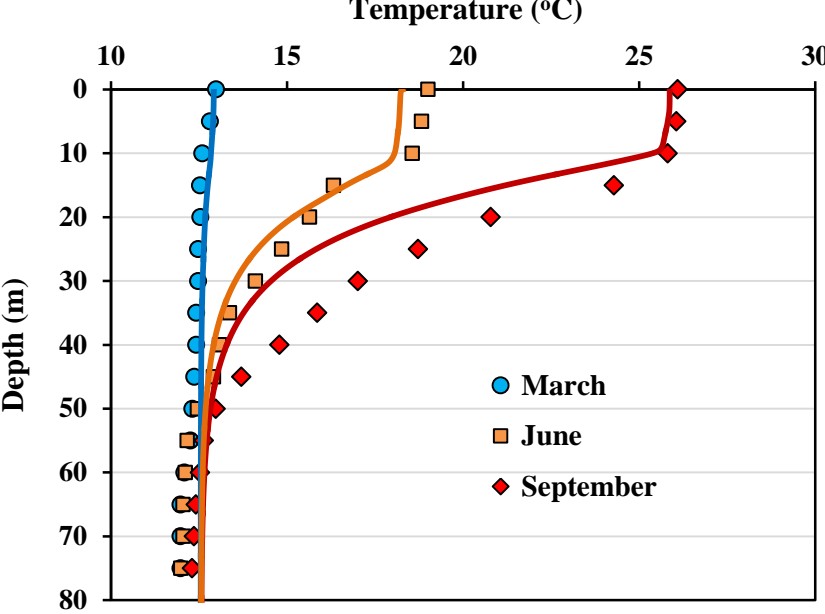

**Figure 5: Model predictions (lines) and measurements (symbols) (Moreo and Swancar, 2013) of mean monthly vertical temperature profiles in Lake Mead; modeling results were obtained using meteorological data measured at Lake Mead assuming radiation absorption at surface ($\beta$) and attenuation coefficient ($\eta$) of 0.3 and 0.1, respectively.**

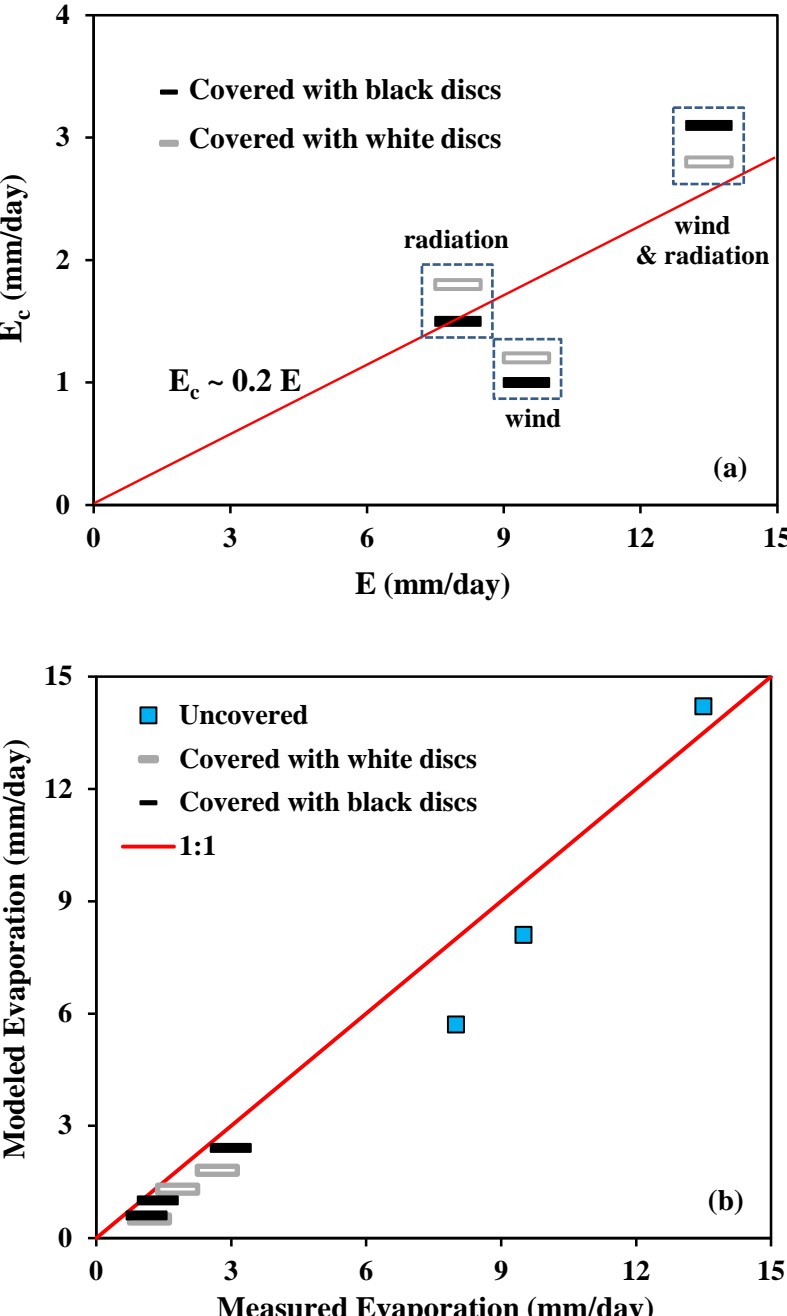

**Figure 6: (a) Laboratory results of evaporative loss from a small water basin covered with floating discs of 0.2 m diameter (see Figure 4); the ratio between evaporation from covered and uncovered reservoir is about 0.2, corresponding to suppression efficiency of about 80%; (b) modeled vs. measured mean evaporation rate from uncovered and covered basin for different forcing marked in (a).**

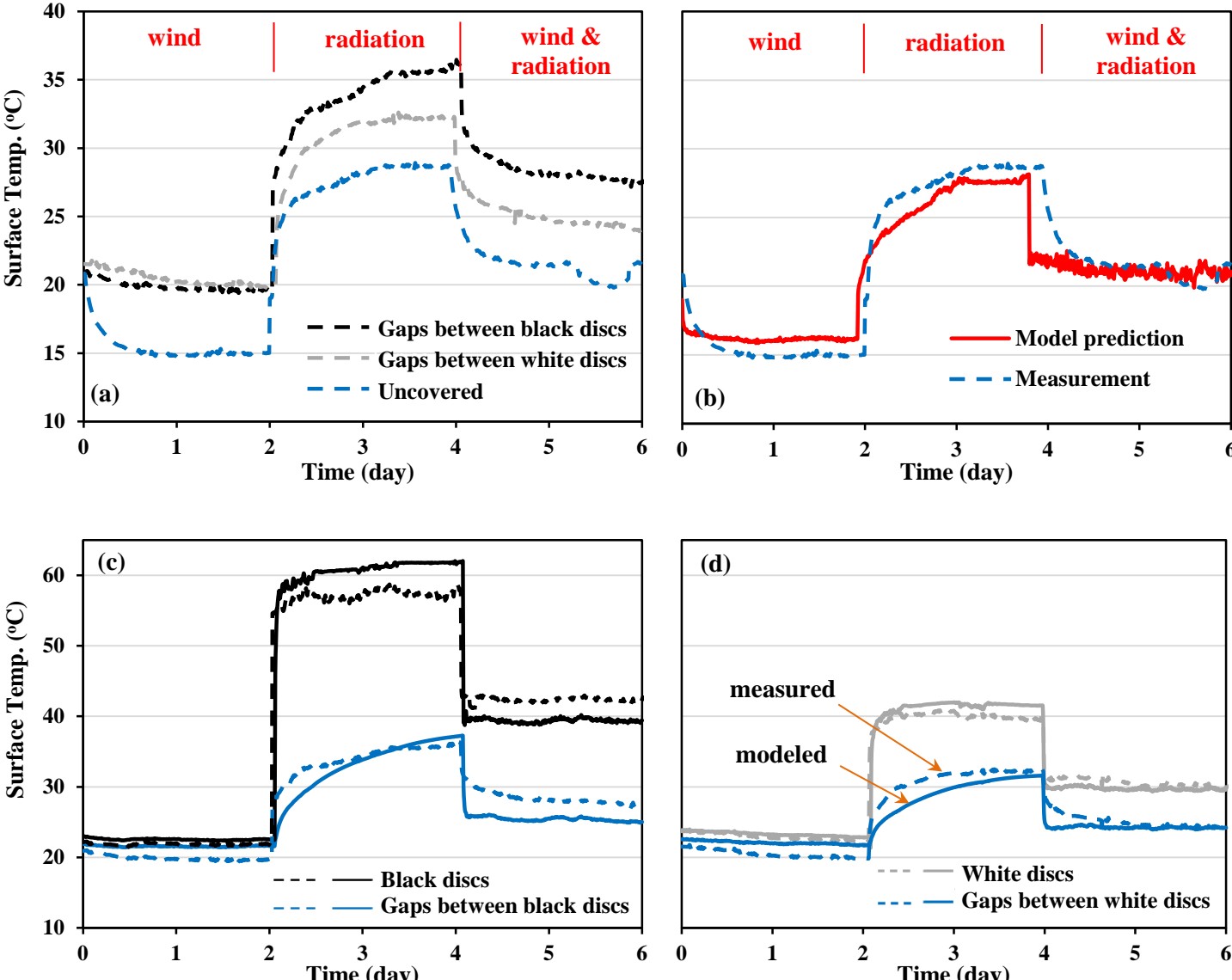

Figure 7: Measured and simulated surface temperature dynamics in lab scale water basin. (a) Surface temperature of uncovered water basin and gaps between white and black discs obtained from IR measurements; (b) model prediction and IR measurements of uncovered water basin surface temperature; (c & d) model predictions (solid lines) and IR measurements (dashed lines) of covers and gaps surface temperature for basin covered with black and white discs, respectively.

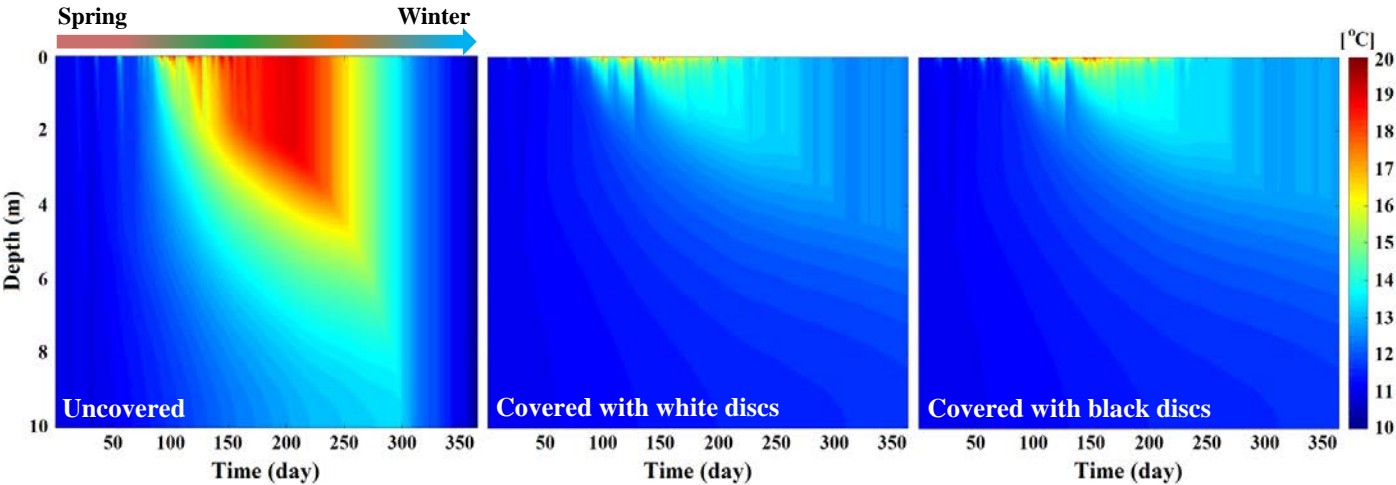

**Figure 8: Modeling the effect of surface coverage on mean daily temperature in a hypothetical reservoir with 10 m depth using meteorological data (European Fluxes Database) in Majadas, Spain (March 2004 to March 2005); the reservoir was covered using white and black Styrofoam discs (diameter: 0.2 m and height: 0.02 m) that provide 0.91 coverage of the reservoir surface. A uniform vertical temperature at 11 °C was assumed as the initial condition, and the bottom boundary condition was set to zero heat flux.**

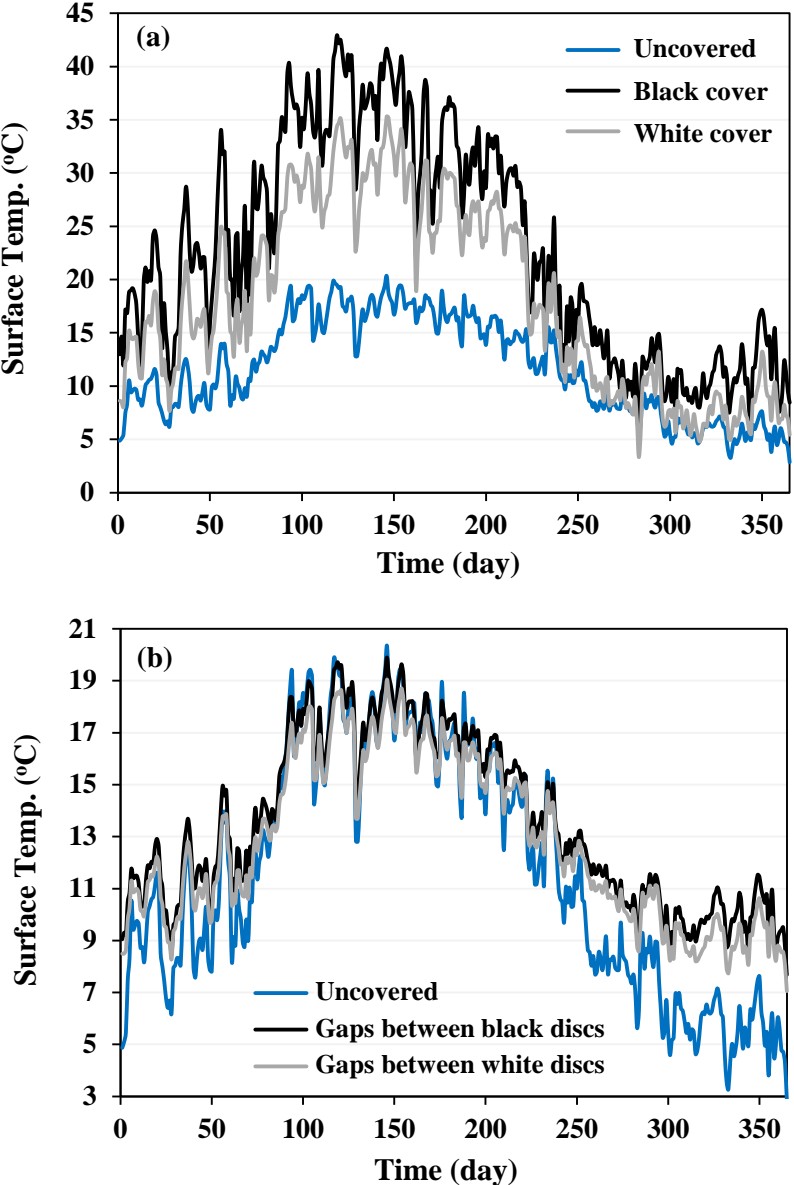

**Figure 9: (a) The evolution of temperature on the top surface of floating discs and on the surface of uncovered reservoir; (b) comparing surface water temperature of the uncovered reservoir and of water gaps between floating elements. The plots show simulation results for a hypothetical reservoir in Majadas (Spain).**

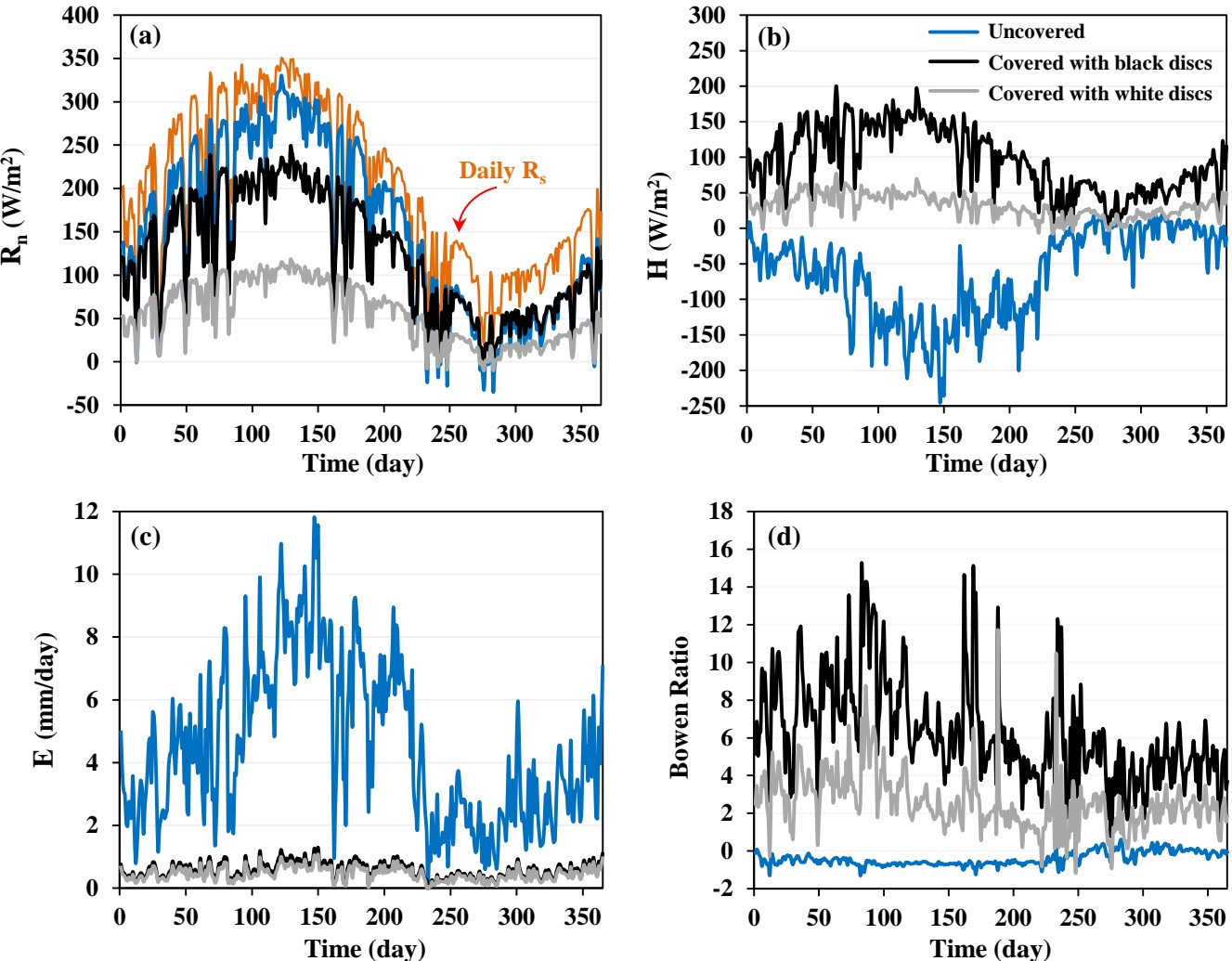

**Figure 10: Model estimates for the evolution of net radiation (a), sensible heat flux (b), evaporation rate (c), and Bowen ratio (d) for uncovered and partially covered reservoir with black and white Styrofoam discs for the meteorological data from Majadas, Spain (March 2004 to March 2005). Mean daily incoming solar radiation is marked in (a).**

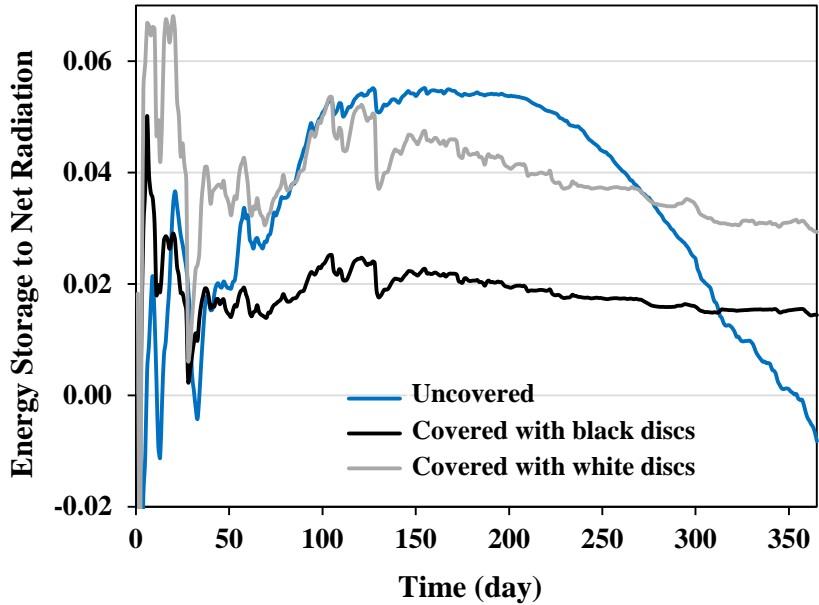

**Figure 11: Model estimates of changes in the ratio of energy storage in the water body to the net radiative flux at the surface of uncovered and partially covered hypothetical 10 m deep reservoir (Majadas, Spain). The heat storage is calculated relative to reference state at the beginning of the water year. Following an equilibration period, the ratio follows the annual variations in the radiative flux for the uncovered reservoir, whereas for the partially covered surface, the ratio remains nearly constant.**

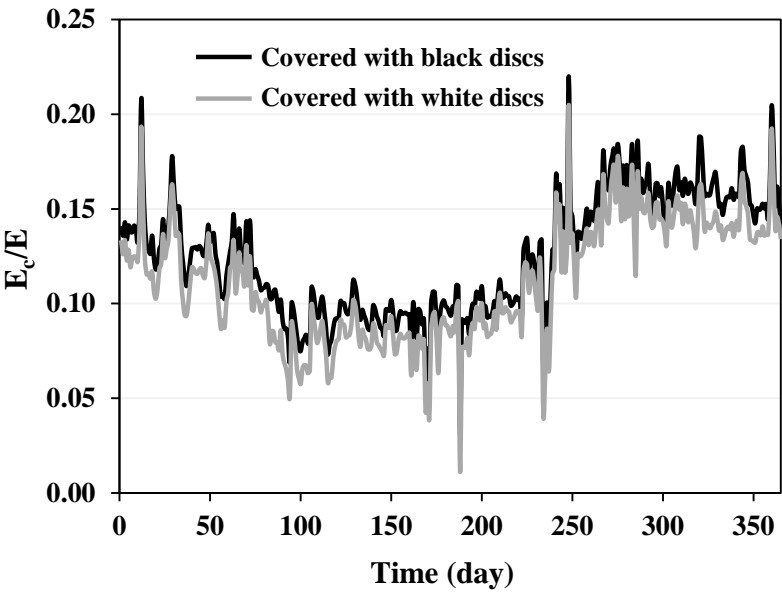

**Figure 12: The ratio of evaporation from covered ($E_c$) to uncovered water reservoir ($E$) representing evaporation suppression efficiency of floating elements (for the meteorological conditions in Majadas, Spain from March 2004 to March 2005).**

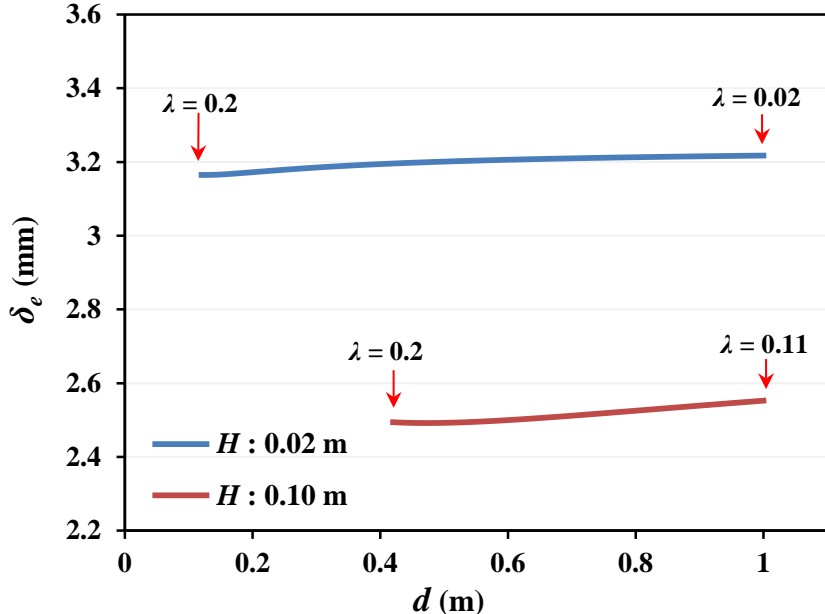

**Figure A1: Variation of effective boundary layer thickness with disc diameter (*d*) for different disc heights (*H*) at wind speed of 1 m/s and surface coverage of 0.91 (dense packing). The increase of $\lambda$ more than 0.2 forms relatively thick boundary layer in the order of disc height. For *U*=1 m/s, the boundary layer thickness over uncovered surface is calculated as 3.2 mm based on Haghighi and Or (2013).**

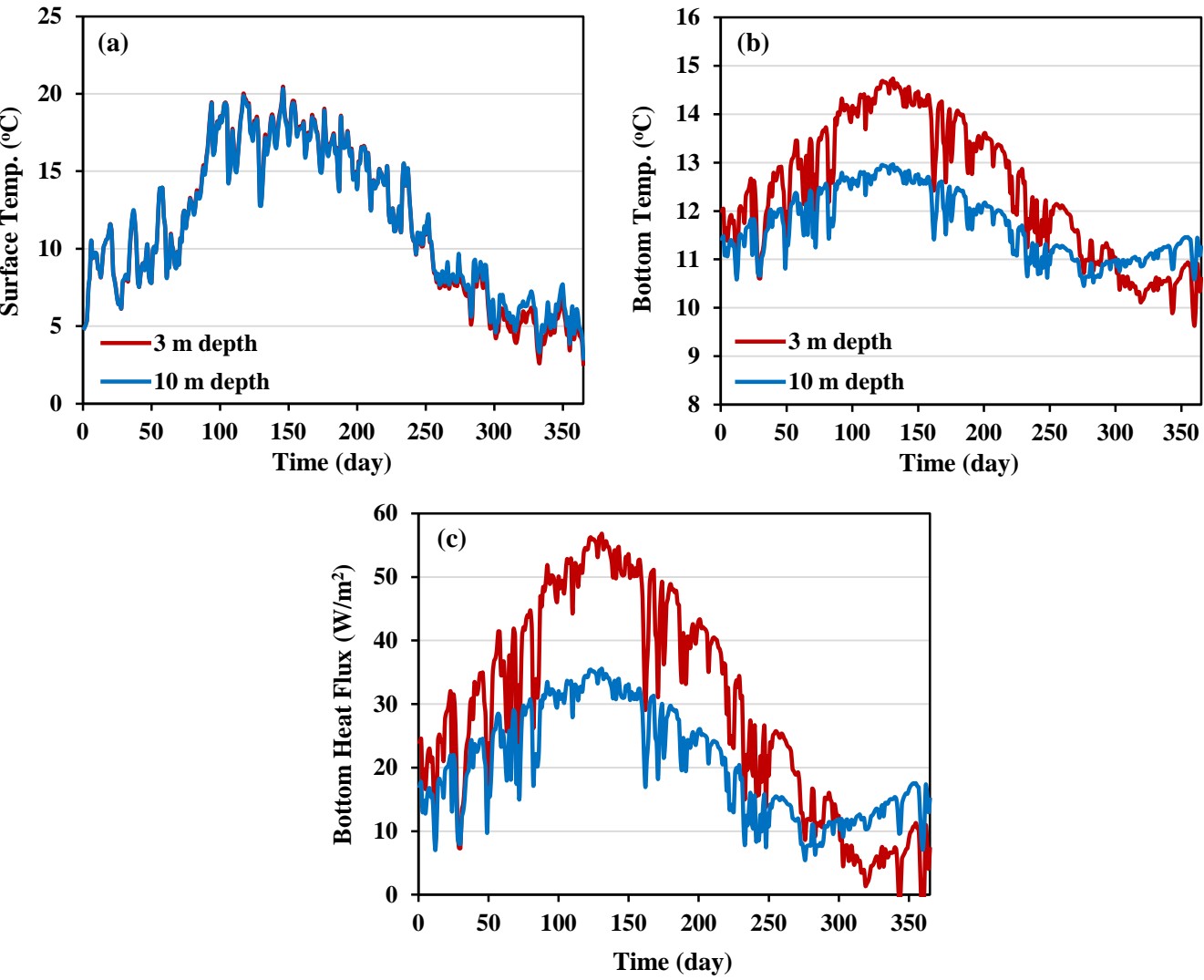

**Figure B1: The effect of reservoir depth on surface (a) and bottom (b) temperature of the uncovered reservoir considering bottom heat flux towards the underlining soil layer (c);** $T_{sZ}$ **was assumed at 10 °C.**