# Peer review of "Evaporation suppression and energy balance of water reservoirs covered with selfassembling floating elements"

_Hydrology and Earth System Sciences, 2017_

## Referee Comment (RC1) · Anonymous Referee #1 · 30 Aug 2017

This manuscript describes the formulation of a numerical model to describe the water quality behaviour of a reservoir covered in floating, evaporation-suppressing elements.

The work undertaken by the authors appears to be novel as I am not aware of previous attempts to quantify numerically the change in fluxes from a reservoir covered in evaporation-suppressing elements.

However, I do have some serious concerns about the manuscript that would need to be addressed before I could support publication.

First, at best, this investigation is speculative as there is no verification data presented to show whether or not is predictions are robust.

[Figure]

At the simplest level, it would be anticipated that if a reservoir was covered in such a way as to prevent the penetration of electromagnetic radiation into its surface by elements of low thermal conductivity, the surface thermal forcing would be reduced. Therefore the outcomes shown in Figure 6 are not surprising.

The authors have elected to use meteorological data that does not appear to have been gathered in the vicinity of a reservoir and apply it to a "hypothetical" reservoir.

The key question is the degree to which the predictions are reliable and the authors have not addressed this question.

As stated on page 11, the authors do have access to a model reservoir that is described in Appendix A. It is difficult to comprehend why they have not verified their model for a system where they were able to make measurements.

At present, I find it difficult to support publication of this manuscript given the degree of speculation in the modelling and that the authors already possess data that could be used to verify it, at least on a small scale.

Secondly, there are considerable inconsistencies in the formulation of the numerical model.

Equation (4) is the conventional expression for stress transfer at an uncovered surface in the absence of wind wave growth. On page 11, line 13 we are told that u*a has been determined from bluff body theory. There is no discussion of the merits of combining these characterisations when their underlying assumptions are clearly at odds.

Also, in Figure 2, the authors invoke a conventional approach to the numerical modelling surface mixing of reservoirs which encapsulates unstable convection due to surface cooling. However, such an approach is unreliable in terms of heat fluxes and the authors' own observations with their infrared camera should show. Certainly the longstanding work by Andy Jessup and his collaborators have revealed very different behaviour of the surface skin (responsible for radiant heat from the surface) from that

of the bulk.

Given these significant inconsistencies, I cannot support publication of the manuscript in its present form.

To progress the speculation to estimating evaporation suppression or oxygen transfer in the absence of any measurements is unjustified in my view. Given the degree of speculation, I cannot support publication of these results until independent verification is available.

In summary, the manuscript has considerable inconsistencies as well as a high degree of speculation. Given these significant weaknesses, I cannot support its publication until these concerns are addresses.

---

## Author Comment (AC1) · 6 Oct 2017

**Response to Reviewer # 1 Comments (manuscript # hess-2017-415):**

**"Evaporation suppression and energy balance of water reservoirs covered with self-assembling floating elements"**

**Milad Aminzadeh, Peter Lehmann, and Dani Or**

Dear Editor,

We greatly appreciate the constructive and insightful comments made by reviewer # 1. In the following, we address the main concerns raised by reviewer regarding a) the model validation, b) stress transfer at a covered surface, and c) surface thermal mixing.

**Reviewer comment:** *First, at best, this investigation is speculative as there is no verification data presented to show whether or not is predictions are robust. At the simplest level, it would be anticipated that if a reservoir was covered in such a way as to prevent the penetration of electromagnetic radiation into its surface by elements of low thermal conductivity, the surface thermal forcing would be reduced. Therefore the outcomes shown in Figure 6 are not surprising. The authors have elected to use meteorological data that does not appear to have been gathered in the vicinity of a reservoir and apply it to a "hypothetical" reservoir. The key question is the degree to which the predictions are reliable and the authors have not addressed this question. As stated on page 11, the authors do have access to a model reservoir that is described in Appendix A. It is difficult to comprehend why they have not verified their model for a system where they were able to make measurements.*

**Reply:** The reviewer is correct that certain aspects pertaining to covered "reservoir scale" predictions remain tentative and would require confirmation using reservoir scale data (the lack of publically available data from covered reservoirs for model validation was acknowledged on page 12, line 19). However, we have tested key aspects of the model using information from uncovered reservoir response to natural conditions (USGS Lake Mead data – see Figure 4) to establish a reference for evaluating how floating covers would affect the energy balance of a covered reservoir. We also incorporated insights from laboratory experiments using floating covers in a "small basin" from our lab at ETH Zurich. Laboratory evaporation experiments using 1.44 m$^2$ basin and 0.16 m depth (as depicted in Figure A) were in good agreement with model predictions for evaporation suppression and effects of cover color (as mentioned in page 15, line 3). Clearly, aspects related to vertical temperature profiles and mixing of covered reservoirs under natural atmospheric conditions would not be captured in such a shallow basin in the lab. In summary, the application of sound physical principles for modeling of evaporation suppression, detailed model validation for uncovered reservoir using multi-seasons measured data, and (limited) insights from laboratory experiments for covered and uncovered basins, render the investigation not so "speculative". Hence, the investigation offers quantitative

predictions for the effects of covers (that indeed do not contradict intuition, except the "surprising" result that white and black covers were equally effective in suppressing evaporation).

[Figure]

Figure A: The water basin with 1.2×1.2 m² surface area and 0.16 m depth in our laboratory; uncovered (left) and covered with black discs (right)

**Reviewer comment:** *Equation (4) is the conventional expression for stress transfer at an uncovered surface in the absence of wind wave growth. On page 11, line 13 we are told that u\*ₐ has been determined from bluff body theory. There is no discussion of the merits of combining these characterizations when their underlying assumptions are clearly at odds.*

**Reply:** The derivations are motivated by the paucity of expressions for aerodynamic interactions of airflows with floating elements on water surfaces (in contrast, numerous studies have addressed interactions with wavy or bluff body covered solid surfaces). The situation is even more complicated considering phase change where heat and mass transfers at evaporating water surfaces potentially affect aerodynamic interactions over such partially covered water surfaces.

The eddy thermal diffusivity in the vertical temperature equation (Eq. 1) was based on a relatively simple and physically-based formulation of Henderson-Seller [1985] (Eq. 3) that expresses eddy diffusivity as a function of friction velocity at the water surface. Note that Eq. (4) emerges from equality of shear stresses at an interface. We invoked a well-established theory of drag partitioning over rough surfaces developed by Shao and Yang [2008] and Nepf [2012] that has been recently evaluated by Haghighi and Or [2015] to quantify the friction velocity ($u_a^*$) of air and define the friction velocity at the water surface ($u_s^*$) based on Eq. (4). Furthermore, we tested this representation by considering the boundary layer thickness (a function of $u_a^*$ [Haghighi and Or, 2015]) obtained from direct measurements of mass loss from our water basin covered with floating discs and simulations using COMSOL. Details of aerodynamic interactions between airflows and floating cover elements are key to evaluation evaporation suppression and thermal effects in covered reservoirs and deserve specially designed studies (beyond the scope of the present work). Nevertheless, we provide additional details on the friction velocity and boundary layer thickness in Appendix B.

**Reviewer comment:** *In Figure 2, the authors invoke a conventional approach to the numerical modelling surface mixing of reservoirs which encapsulates unstable convection due to surface cooling. However, such an approach is unreliable in terms of heat fluxes and the authors' own observations with their infrared camera should show. Certainly the longstanding work by Andy Jessup and his collaborators have revealed very different behavior of the surface skin (responsible for radiant heat from the surface) from that of the bulk.*

**Reply:** We thank the reviewer for raising this point. Equation (1) is a generic differential equation for describing vertical temperature profiles either in a water body or in a solid slab (with $D_w = 0$). What differentiates the solutions for these two cases is vertical mixings in water body triggered by thermal/density instabilities (e.g., a cold layer of water due to evaporative cooling overlying warmer water below). Such mixing processes are triggered at small scales diurnally (due to evaporative cooling at the surface), or seasonally where subsurface heat accumulation raises to the surface and unifies the vertical temperature profile in a reservoir (either Monomictic or Dimictic reservoirs). Note that the simple vertical mixing approach of Dake and Harleman [1969] preserves the energy balance of the reservoir.

We fully agree with the reviewer that surface heat fluxes vary by imposing such mixing scenario. We thus imposed the vertical mixing on the "mean daily" temperature profile providing the initial condition at the beginning of next day. This step prevents transfer of heat towards the bottom of the water body (such as would happen in a solid slab). Consequently, the "instantaneous" values of surface temperature and surface heat fluxes are obtained directly from the temperature equation with surface boundary conditions represented in Eq. (8) or (12), hence unaffected by surface thermal mixing as depicted in Figure 2. This can be seen, for example, by comparing winter surface temperature of uncovered reservoir in Figures 5 and 6b. The good agreement between model predictions of vertical temperature profile in Lake Mead and measurements (Figure 4) further confirms our modeling approach based on Dake and Harleman [1969] without affecting the calculation of surface temperature and thus surface heat fluxes represented in Table 2.

We thank again the reviewer for many helpful comments and hope the Editor finds the clarifications satisfactory.

Sincerely,
Milad Aminzadeh, Peter Lehmann, and Dani Or

---

## Referee Comment (RC2) · Anonymous Referee #2 · 4 Nov 2017

Review of "Evaporation suppression and energy balance of water reservoirs covered with self-assembling floating elements" By Milad Aminzaheh, Peter Lehmann, Dani Or Recommendation: Accept with some revisions General Comment: This is a well-written and presented article. It provides a relatively simple but surprisingly comprehensive theoretical and physical basis of evaporation suppression from simple, shallow reservoirs from which more detailed work can emerge. It does this by comparing models of an uncovered reservoir to ones covered by white and black circular discs. A 1-D, column approach was used. I wondered why triangular covers were not considered as they have the potential of having no gaps between them (or much smaller ones than a disc).

The paper could be well served by articulating right at the outset the methodology you use. This is how I perceive it (from reading p. 11): 1. Calculation of evaporation reduction due to discs 2. Effect of heat balance of the discs on water column, the primary evaporation reduction element 3. Effect of heat balance of the gaps between discs on water column, including conduction from disc to water. 4. Effect of the increase of gap water surface temperature due to 2 and 3.

Advection of (likely) colder water into the column was brought up in a discussion of managed input vs output for the reservoir but non-advective heat transfer was only considered for the bottom of the column. What about the four sides (can assume a simple soil temperature profile)?

The diffusivity coefficient, D, did not appear to include any internal dynamics such as non-linear and/or breaking waves, which would likely increase it. The authors might consider such inclusion for completeness. Although, I must admit, internal motions in such a shallow reservoir would not be very large or complex. However, I am not aware of any observations of internal motions in shallow reservoirs and there are few for larger, deeper ones (with bottom topography forcing the wave motion). Managed releases would exacerbate wave activity.

It appeared implicitly assumed that the water was not turbid, a rare condition in most reservoirs. A short discussion of the effect of turbidity on the columnar distribution of heat would enhance the work and provide an avenue for further theoretical work.

While the amount of open water subject to heating is small in this study, for completeness at least a nod to the Claeusius-Clapeyron relationship should be noted (and, I guess, dismissed). It had a major impact on the "failure" of monomolecular layer cover evaporation suppression in the famous Lake Hefner (Oklahoma, USA) Evaporation Reduction Experiment in 1967 (Bean and Florey, 1968, Water Resources Res., 4, 206-208; also notes an evaporation reduction of about 60%) because the water warmed up when evaporation was reduced. Wind removed the layer, exposing the warm water,
which then had higher evaporation due to the warmer water resulting in a net loss.

One must ask, though, if the general design of these small, shallow reservoirs, a given in this work, is a good beginning. So while physical intervention, as discussed here, is important, the general design of a reservoir is equally important.

An important metric, the mean depth, $D = V/A$, where V is the reservoir volume and A, its surface area was not discussed. An efficient reservoir would be one where V is large and A is small resulting in a large value of D; in other words a cylinder will evaporate less than a bowl of the same volume. In this case $3m < D < 10m$ was considered. This is very shallow, implying a rapid response of reservoir heat content to varying atmospheric forcing; in other words the surface temperature, the main driver of the evaporative process, responds rapidly to latent and sensible heat transfer as well as the mean temperature of the volume. There is little phase lag between the near surface heat balance and interior heat balance; both will closely follow the daily average air temperature and net radiation input.

In a deeper reservoir, Lake Mead was used where D is 165, there is a considerable phase lag in the diurnal and seasonal variations of surface versus interior temperature. For instance, in summer daytime air temperature will likely exceed the water temperature; a stable situation resulting in reduced evaporation especially in windless conditions. The reverse is true at night, when water temperatures are likely warmer than air temperature. Since during summer mid-latitude daylight hours substantially exceed nighttime hours so the lower evaporation during the day will dominate. In Fall, surface temperature will decrease due to lower insolation amount and duration, but will this will likely be mitigated by heat transfer into the surface layer by relatively warmer water in the interior resulting in relatively warmer surface temperature than air temperature throughout the day resulting in potentially more evaporation in that season (and Winter) compared to summer. The results shown in this article do not support this heuristic argument. However, eddy correlation observations over a period of years over Lake Superior (Blanken, P. et al., 2011, J. Great Lakes Res., 37, 707-716) show

this nicely.

Last, I recall talking with a farmer who was the leader of a ditch company that managed a small reservoir as assumed here. He was very interested in estimating evaporation and, of course, suppressing it with some sort of cover as described here. I asked him if he had planted a wind break on the windward side. He was stunned and said he had not thought of it. So I said: "But you thought of it for your fields and that isn't open water. Furthermore, it would be a good use of otherwise "lost" leakage to ground water." So while I understand this windbreak approach and the consideration of internal boundary layers formed by changes in surface friction is not conducive to such a study as outlined here, I feel a theoretical approach to these aspects of real world reservoirs would be worthwhile in the search for low-impact geoengineering of simple reservoirs. This group obviously have the tools and expertise.

Specific Comments using page, line, equation, figure and table numbers

p. 2, l. 15: I believe the recent use of black balls in a Los Angeles reservoir was not aimed at evaporation reduction but the reduction of toxic algae blooms. I think Israeli engineers have used white ping-pong like balls to reduce evaporation in test reservoirs (don't have a reference).

p. 7, eq. 6a: Please check for references for some of these empirical relationships. Some equations are referenced, some not.

p. 7, eq. 6b: some readers will not recognize the Brundt-Viasala relationship, which carries some restrictive assumptions with it. Interestingly on a windless or low wind day, this might be more likely during the day and convective mixing, as noted in this work, which is more likely at night when surface temperature might be lower than temperatures below.

p. 7, l. 12-13: Do you have a reference for the assumption?

p. 8, eq. 8: explain why you use C for vapor concentration instead of the more recog-

nizable q, specific humidity.

p. 8, l. 15, Fig. 2: is this the heavy dashed line in the Figure? It needs to be explained.

p. 10, l. 13-14: jargon alert! "three-dimensional vapor shells" Show or explain further. Also what is meant by "lateral spacing"? Perhaps you can show these in Fig. 3b.

p. 12, l. 12: Consider "Given the simplifying assumptions, the model overestimates. . ..".

P. 13, Fig. 5: Comment on the slow uptake of heat in Spring (cold water/warm air) vs rapid decrease in Fall (warm water/cold air) to add confidence in the model. You might find observational evidence to back up a heuristic argument: surface layer more stable in Spring, more convective in Fall.

p. 13, l. 18-19 "..demonstrate . . . a much colder reservoir." This is an impressive modeling result and should be tested by a field experiment. Is one being considered?

p. 17, Section 3.3, Ecological considerations: Reservoirs, even small, simple ones as assumed here, while not likely used for recreation, can be important to migratory birds and other wildlife as well as aquatic life in the reservoir (which often provide food for wildlife visiting the reservoir, extending the ecological boundary). Discs, as described here, will inhibit access for wildlife. That should at least be mentioned along with the impossibility of modeling it. Although, for any future work, you might consider entraining a wildlife expert who might.

p. 18, Section 3.4, Costs and water savings: A nice summary. Have an economist vet it, if you haven't. I especially liked the last sentence. You might mention water scarcity as a conflict enhancer as described recently by Tom Friedman, a well-known columnist for the New York Times. So efficient storage of water becomes political.

p. 22, l. 10-18: You should mention an important effect of the discs I did not see in the paper but suggested in this discussion; the appearance of waves, breaking waves, and spray as wind increases (threshold $\sim$ 6 mps). This radically changes the situation in open, uncovered water and greatly increases the evaporation; modeling this effect
is still elusive though a check of hurricane boundary layer modeling may provide some insights.

P. 23, Eq. B3: Is this correct? Should it be $\lambda = D/H$ to be dimensionless as described later in the Appendix?

p. 24, l. 13-15: 3 to 10 m is not enough depth variation. Note that Rn, H, and E are essentially constant and heat storage decreased by 27% for the 10m depth. Can you show the "index" Ta – Tw for the two depths? I predict they will be nearly the same.

Figure 1: I'd replace that with a Google Earth picture of the Front Range (eastward) of Colorado which is dotted with small reservoirs to show how ubiquitous they are. Using the area tool on the USGS National Map Viewer you could show that the combined area of these "small" reservoirs approximate that of major reservoirs in the Colorado River Basin system.

Figure 2: Those two hatched areas do not look equal to me. Explain the dashed line.

Figure 3: What is the red triangle on the far left side? You've labeled the down arrows to the far left and right, what is the label for the one between them? What does the expression below fc represent?

Figure 4: "assumed"? Be honest, wasn't it "tuned"? Were "$\eta$" and "$\beta$" observed?

Figure 7: I think 7c is a result of the shallowness of the reservoirs you are modeling. They are like an evaporation pan which has a similar trace with respect to season. Deeper reservoirs show a maximum in Fall/Winter and a minimum in Summer for good reasons. Dew forms on the surface of Lake Superior in summer!! I've witnessed explosive evaporation events associated with reservoir overturning in mid-winter with air temperature of -12C.

---

## Author Comment (AC2) · 21 Nov 2017

**Response to Reviewer # 2 Comments (manuscript # hess-2017-415):**

**"Evaporation suppression and energy balance of water reservoirs covered with self-assembling floating elements"**

**Milad Aminzadeh, Peter Lehmann, and Dani Or**

Dear Editor,

We greatly appreciate the constructive and insightful comments made by reviewer # 2. In the following, we address the comments and concerns raised by the reviewer.

**Reviewer comment:** *This is a well written and presented article. It provides a relatively simple but surprisingly comprehensive theoretical and physical basis of evaporation suppression from simple, shallow reservoirs from which more detailed work can emerge. It does this by comparing models of an uncovered reservoir to ones covered by white and black circular discs. A 1-D, column approach was used. I wondered why triangular covers were not considered as they have the potential of having no gaps between them (or much smaller ones than a disc).*

**Reply:** We thank the reviewer for the efforts and for the many insightful comments. As mentioned in Page 10 (L 10-16), the aerodynamic resistance for vapor flux from water gaps forming between cover elements is governed by the combined effects of gap size ($a_g$), boundary layer thickness ($\delta$) and the lateral spacing between neighboring gaps. For very small gaps formed by polygonal covers, gap sizes could become smaller than the boundary layer thickness (the ratio of $a_g / \delta$ smaller than 1). This case may yield evaporation enhancement disproportional to size of the gap according to Eq. (13) [Schlunder, 1988; Shahraeeni et al., 2012]. In addition, for certain applications of multiuse reservoirs, water gaps formed between spherical or cylindrical covers allow light penetration and provide surfaces oxygen transfer both play important roles in ecological aspects of the water body.

**Reviewer comment:** *The paper could be well served by articulating right at the outset the methodology you use. This is how I perceive it (from reading p. 11): 1. Calculation of evaporation reduction due to discs; 2. Effect of heat balance of the discs on water column, the primary evaporation reduction element; 3. Effect of heat balance of the gaps between discs on water column, including conduction from disc to water; 4. Effect of the increase of gap water surface temperature due to 2 and 3.*

**Reply:** We thank the reviewer for the suggestion and we will provide a summary of the main steps and methodology in the revised manuscript.

**Reviewer comment:** *Advection of (likely) colder water into the column was brought up in a discussion of managed input vs output for the reservoir but non-advective heat transfer was only considered for the bottom of the column. What about the four sides (can assume a simple soil temperature profile)?*

**Reply:** Clearly, for small reservoirs lateral heat exchange with the water body could be important in the energy balance. At this stage we seek to establish a simple 1-D model for reservoirs where vertical temperature profile and surface heat fluxes dominate the response in the presence of floating elements, we thus neglect lateral heat transfers of the reservoir assuming that the side area of the reservoir is

small relative to its depth (as is likely in many shallow reservoirs). Following this comment, we will explicitly mention this simplification in the revised manuscript (to also consider this aspect in applications to small ponds).

**Reviewer comment:** *The diffusivity coefficient, D, did not appear to include any internal dynamics such as non-linear and/or breaking waves, which would likely increase it. The authors might consider such inclusion for completeness. Although, I must admit, internal motions in such a shallow reservoir would not be very large or complex. However, I am not aware of any observations of internal motions in shallow reservoirs and there are few for larger, deeper ones (with bottom topography forcing the wave motion). Managed releases would exacerbate wave activity.*

**Reply:** We note that some of the internal motions in deep reservoirs are attributed to the onset of thermal instabilities as included in the model representation (Eq. 10); additionally, effects of wind friction velocity are explicit in the (nonlinear) formulation of the eddy diffusivity in Eq. (3). Clearly, inflows-outflows, bottom topography, and breaking waves would enhance mixing and thus modify effective eddy diffusivity. However, keeping with the simple 1-D formulation of Henderson-Seller [1985], we retain surface interactions of eddy diffusivity with wind (that is likely to be altered in the presence of the floating cover!). Following the reviewer comment, we will explain these aspects in the revised manuscript.

**Reviewer comment:** *It appeared implicitly assumed that the water was not turbid, a rare condition in most reservoirs. A short discussion of the effect of turbidity on the columnar distribution of heat would enhance the work and provide an avenue for further theoretical work.*

**Reply:** We thank the reviewer for raising this point; we have considered parameterization of light penetration and various radiative effects in Eq. (2), but we will add a short discussion of this aspect in the revised manuscript (with common values and ramifications).

**Reviewer comment:** *While the amount of open water subject to heating is small in this study, for completeness at least a nod to the Clausius-Clapeyron relationship should be noted (and, I guess, dismissed). It had a major impact on the "failure" of monomolecular layer cover evaporation suppression in the famous Lake Hefner (Oklahoma, USA) Evaporation Reduction Experiment in 1967 (Bean and Florey, 1968, Water Resources Res., 4, 206- 208; also notes an evaporation reduction of about 60%) because the water warmed up when evaporation was reduced. Wind removed the layer, exposing the warm water, which then had higher evaporation due to the warmer water resulting in a net loss.*

**Reply:** We thank the reviewer for raising this important point. We include the Clausius-Clapeyron relationship in the representation of evaporative flux based on the "saturated vapor concentration" that is a function of surface temperature (e.g., see Eq. 8). The difference from the cases mentioned, is with the energy balance over the covers either via albedo reflection (white covers) or sensible heat exchange (black covers) with minimal net heat flux to the surface (laboratory experiments). We will add a discussion of the potential nonlinear evaporation enhancement effects as a function of surface water temperature.

**Reviewer comment:** *An important metric, the mean depth, D = V/A, where V is the reservoir volume and A, its surface area was not discussed. An efficient reservoir would be one where V is large and A is small resulting in a large value of D; in other words a cylinder will evaporate less than a bowl of the same volume. In this case 3m < D < 10m was considered. This is very shallow, implying a rapid response of reservoir heat content to varying atmospheric forcing; in other words the surface temperature, the main driver of the evaporative process, responds rapidly to latent and sensible heat*

*transfer as well as the mean temperature of the volume. There is little phase lag between the near surface heat balance and interior heat balance; both will closely follow the daily average air temperature and net radiation input.*

*In a deeper reservoir, Lake Mead was used where D is 165, there is a considerable phase lag in the diurnal and seasonal variations of surface versus interior temperature. For instance, in summer daytime air temperature will likely exceed the water temperature; a stable situation resulting in reduced evaporation especially in windless conditions. The reverse is true at night, when water temperatures are likely warmer than air temperature. Since during summer mid-latitude daylight hours substantially exceed nighttime hours so the lower evaporation during the day will dominate. In Fall, surface temperature will decrease due to lower insolation amount and duration, but will this will likely be mitigated by heat transfer into the surface layer by relatively warmer water in the interior resulting in relatively warmer surface temperature than air temperature throughout the day resulting in potentially more evaporation in that season (and Winter) compared to summer. The results shown in this article do not support this heuristic argument. However, eddy correlation observations over a period of years over Lake Superior (Blanken, P. et al., 2011, J. Great Lakes Res., 37, 707-716) show this nicely.*

**Reply:** We note that the present model was developed with relatively shallow reservoirs in mind (i.e., depth<10 m). As pointed out by the reviewer, the response of such shallow reservoirs to atmospheric condition is rapid and the time lag between surface and interior heat balances is relatively short. A comparison with data from Lake Mead (a relatively deep reservoir), enabled testing of key aspects of the model towards establishing a reference uncovered surface for evaluating effects of floating covers on the energy balance (the main objective of the present study). We agree with the reviewer that in the presence of phase lags and multiple mixing (e.g., dimictic reservoirs), the evaporative flux could be affected and even leading to enhanced evaporative losses during the winter. We point out however, that the monthly evaporation data from Lake Mead support the results of higher evaporative losses during summer even for such a deep reservoir (the Figure A below). We will add a discussion of the subject (and potential deviations from the assumptions) in the revised manuscript.

[Figure]

Figure A. Comparing model estimations of evaporative loss from Lake Mead with measurements demonstrating that evaporative losses during summer are dominant.

**Reviewer comment:** *Last, I recall talking with a farmer who was the leader of a ditch company that managed a small reservoir as assumed here. He was very interested in estimating evaporation and, of course, suppressing it with some sort of cover as described here. I asked him if he had planted a wind break on the windward side. He was stunned and said he had not thought of it. So I said: "But you thought of it for your fields and that isn't open water. Furthermore, it would be a good use of otherwise "lost" leakage to ground water." So while I understand this windbreak approach and the consideration of internal boundary layers formed by changes in surface friction is not conducive to such a study as outlined here, I feel a theoretical approach to these aspects of real world reservoirs would be worthwhile in the search for low-impact geoengineering of simple reservoirs. This group obviously has the tools and expertise.*

**Reply:** We thank the reviewer for sharing these insights; we also consider aspects of internal boundary layer and its impact on local heat and mass transfer processes key to the efficiency of the cover. We thus plan to further investigate such nuanced aspects in the next steps of this ongoing project to provide a comprehensive framework, including both physical and ecological aspects for design and management of (optimal) floating elements.

**Minor comments**

- *p. 2, l. 15: I believe the recent use of black balls in a Los Angeles reservoir was not aimed at evaporation reduction but the reduction of toxic algae blooms. I think Israeli engineers have used white ping-pong like balls to reduce evaporation in test reservoirs (don't have a reference).*

  **Reply:** The reviewer is right, the initial motivation was suppression of photochemical reactions and evaporation suppression from Los Angeles reservoirs was a secondary goal. Nevertheless, the water saving aspect gained prominence with the lingering drought in California (as reflected in highly publicized media cover: https://www.engadget.com/2016/09/21/the-big-picture-shade-balls-los-angeles-reservoir/)

- *p. 7, eq. 6a: Please check for references for some of these empirical relationships. Some equations are referenced, some not.*

  **Reply:** As mentioned in Page 6, L 9, Eq. (3) and subsequent equations used for quantification of eddy thermal diffusivity are provided by Henderson-Seller [1985]. Following the comment of the reviewer, we will explicitly point it in the revised manuscript.

- *p. 7, eq. 6b: some readers will not recognize the Brundt-Viasala relationship, which carries some restrictive assumptions with it. Interestingly on a windless or low wind day, this might be more likely during the day and convective mixing, as noted in this work, which is more likely at night when surface temperature might be lower than temperatures below.*

  **Reply:** We thank the reviewer for the comment; the Brunt-Viasala relation is part of the stability parameterization of the eddy thermal diffusivity by Henderson-Seller [1985] and plays an important role in the water mixing we consider in this work.

- *p. 7, l. 12-13: Do you have a reference for the assumption?*

  **Reply:** The assumption arises from the continuity of temperature profile at the interface of liquid and solid phases. We will provide appropriate references in the revised manuscript.

- *p. 8, eq. 8: explain why you use C for vapor concentration instead of the more recognizable q, specific humidity.*

  **Reply:** The representation based on the vapor concentration arises from Fickian mass transfer across the air boundary layer (implying dominance of diffusive fluxes [Haghighi et al., 2012]). In any case, we don't expect this to affect the clarity of the analysis as vapor concentration and specific humidity are linked via air density (we will add a comment for the readers more comfortable with $q$).

- *p. 8, l. 15, Fig. 2: is this the heavy dashed line in the Figure? It needs to be explained.*

  **Reply:** It is represented by the solid line denoted as $T_m$, we will remove the heavy dashed line in the revised manuscript to avoid confusion of the readers.

- *p. 10, l. 13-14: jargon alert! "three-dimensional vapor shells" Show or explain further. Also what is meant by "lateral spacing"? Perhaps you can show these in Fig. 3b.*

  **Reply:** The reviewer is right to "alert" of such jargon use. The point here is that for small lateral spacing between neighboring water gaps in the cover (either for covers made up of small elements, or densely punctured plastic cover), the vapor concentration profile resulting over the surface is nearly 1-D and layered (left image below); as spacing increases, the vapor profiles form individual 3-D domes that act to enhance evaporative flux from individual gaps. This is schematically represented in the image below. Following the reviewer comment, we will explain it better in the revised manuscript.

[Figure]

  Figure B. Conceptual image of evolution of vapor shells above individual water gaps with increasing spacing between them.

- *p. 12, l. 12: Consider "Given the simplifying assumptions, the model overestimates: : :.".*

  **Reply:** Thanks, we will amend the sentence in the revised manuscript.

- *P. 13, Fig. 5: Comment on the slow uptake of heat in Spring (cold water/warm air) vs rapid decrease in Fall (warm water/cold air) to add confidence in the model. You might find observational evidence to back up a heuristic argument: surface layer more stable in Spring, more convective in Fall.*

  **Reply:** We appreciate this constructive comment and will provide further discussions on the evolution of temperature profile in the revised manuscript.

- *p. 13, l. 18-19 "..demonstrate : : : a much colder reservoir." This is an impressive modeling result and should be tested by a field experiment. Is one being considered?*

  **Reply:** We are aware that model predictions require confirmation using reservoir scale experiments that are currently unavailable. We note that a colder water body under floating elements was observed in our preliminary lab scale measurements in a small basin ($1.2 \times 1.2 \times 0.16$ m$^3$), and are presently conducting two field scale measurements (in EAWAG

(Switzerland) and Isfahan University of Technology (Iran); see images) to provide the necessary data for model evaluation.

[Figure]

[Figure]

Figure C. Ongoing "field scale" experiments conducted in EAWAG near Zurich using 8 ponds each 14 m² and 1.5 m deep covered with white and black 0.2 m (EVA foam) floating covers including two uncovered control ponds

- *p. 17, Section 3.3, Ecological considerations: Reservoirs, even small, simple ones as assumed here, while not likely used for recreation, can be important to migratory birds and other wildlife as well as aquatic life in the reservoir (which often provide food for wildlife visiting the reservoir, extending the ecological boundary). Discs, as described here, will inhibit access for wildlife. That should at least be mentioned along with the impossibility of modeling it. Although, for any future work, you might consider entraining a wildlife expert who might.*

  **Reply:** The reviewer is right, ecological aspects of covered reservoirs are not limited to aquatic organisms only and additional aspects including birds and wildlife should be considered. Future development of the framework will consider other more nuanced ecological aspects such as optimizing surface coverage to provide required light and oxygen for aquatic life and accessibility to other organisms as pointed out.

- *p. 18, Section 3.4, Costs and water savings: A nice summary. Have an economist vet it, if you haven't. I especially liked the last sentence. You might mention water scarcity as a conflict enhancer as described recently by Tom Friedman, a well-known columnist for the New York Times. So efficient storage of water becomes political.*

  **Reply:** We thank the reviewer for raising this point and will further discuss (the rapidly evolving) economics of water saving in the revised manuscript and in follow up studies.

- *p. 22, l. 10-18: You should mention an important effect of the discs I did not see in the paper but suggested in this discussion; the appearance of waves, breaking waves, and spray as wind increases (threshold _ 6 mps). This radically changes the situation in open, uncovered water and greatly increases the evaporation; modeling this effect is still elusive though a check of hurricane boundary layer modeling may provide some insights.*

  **Reply:** We will discuss the impact of discs on surface waves, shear velocity and potential impacts on the evaporative loss in the revised manuscript (albeit this will be done in a generic fashion due to lack of data and much larger scope than the present study).

- *P. 23, Eq. B3: Is this correct? Should it be $\lambda = D/H$ to be dimensionless as described later in the Appendix?*

**Reply:** Please note that the dimension of $d.H$ is m$^2$ and the parameter $N$ represents number of discs per unit "area" rendering $\lambda$ a dimensionless parameter ($\lambda = N d H$).

- *p. 24, l. 13-15: 3 to 10 m is not enough depth variation. Note that Rn, H, and E are essentially constant and heat storage decreased by 27% for the 10 m depth. Can you show the "index" Ta – Tw for the two depths? I predict they will be nearly the same.*

**Reply:** Note that the focus of the present study is on shallow reservoirs often used for seasonal water storage for domestic, agricultural or industrial use in dry periods. The similarity of surface heat fluxes between shallow and deep scenarios is associated with similar surface temperatures and the effect of depth is thus reflected in specific storage and bottom fluxes. Following the reviewers comment, the following plot depicts the difference between air temperature and mean vertical temperature of the reservoir. Although they look similar, the differences in summer and winter are of the order of 3$^o$C.

[Figure]

Figure D. Variation of $T_a$-$T_w$ for shallow and deep reservoirs. The water temperature ($T_w$) represents the mean vertical temperature of each day.

- *Figure 1: I'd replace that with a Google Earth picture of the Front Range (eastward) of Colorado which is dotted with small reservoirs to show how ubiquitous they are. Using the area tool on the USGS National Map Viewer you could show that the combined area of these "small" reservoirs approximate that of major reservoirs in the Colorado River Basin system.*

**Reply:** Thanks for the suggestion; we will consider using the map of the suggested area in the revised manuscript.

- *Figure 2: Those two hatched areas do not look equal to me. Explain the dashed line.*

**Reply:** Considering Eq. (10), the hatched areas on the left and right hand sides of $T_m$ are the same. As mentioned earlier, we will remove dashed line in the revised manuscript to avoid confusion of potential readers.

- *Figure 3: What is the red triangle on the far left side? You've labeled the down arrows to the far left and right, what is the label for the one between them? What does the expression below fc represent?*

**Reply:** As expressed in the caption of Figure 3, the red triangle with side lengths equal to the disc diameter marks a unit cell that enables calculation of surface coverage based on the geometry as 0.91. The arrow between the down arrows is $q_c$; we will add it to the revised Figure. The expression indicates attenuation of radiative flux in depth; we will better highlight it in the revised manuscript.

- *Figure 4: "assumed"? Be honest, wasn't it "tuned"? Were "η" and "β" observed?*

  **Reply:** To obtain the radiative properties and light attenuation in Lake Mead, we already contacted Dr. Michael Moreo at USGS who is responsible for measurements at the lake; his reply was "*I do not have any subsurface radiation attenuation data. I will say the lake is very clear, and I suspect that radiation penetration is as deep as other very clear lakes*". We thereby estimated the values of $\eta$ and $\beta$ based on the literature data reported for clear water bodies.

- *Figure 7: I think 7c is a result of the shallowness of the reservoirs you are modeling. They are like an evaporation pan which has a similar trace with respect to season. Deeper reservoirs show a maximum in Fall/Winter and a minimum in Summer for good reasons. Dew forms on the surface of Lake Superior in summer!! I've witnessed explosive evaporation events associated with reservoir overturning in mid-winter with air temperature of -12C.*

  **Reply:** Please note that higher evaporative loss during winter is not general as observation in Lake Mead indicates higher evaporation during summer (please see Figure A above). Considering the reviewer's comment, we thus recall that the focus of the present work is on relatively shallow reservoirs whereby evolution of surface fluxes are expected to follow seasonal cycles with higher evaporative loss in summer.

We thank again the reviewer for many helpful comments and hope the Editor finds the clarifications satisfactory.

Sincerely,

Milad Aminzadeh, Peter Lehmann, and Dani Or

---

## Author Response (AR1)

**Response to Reviewers' Comments (manuscript # hess-2017-415):**

**"Evaporation suppression and energy balance of water reservoirs covered with self-assembling floating elements"**

**Milad Aminzadeh, Peter Lehmann, and Dani Or**

Dear Editor,

We greatly appreciate the constructive and many insightful comments made by the Editor and two reviewers. In the following we first address the points raised by the Editor:

*As someone who is familiar with the challenges of evaporation reduction and also the complexities of lake mixing I believe the study has significant merit and in particular is in an area where there is currently limited literature. Bringing a theoretical energy balance/aerodynamic approach to the problem I think will make the study a useful contribution.*

*The reviewers have a somewhat mixed view of the status of the manuscript in its present form, and so I have spent a fair bit of time myself reviewing and assessing the paper. In short, Reviewer 2 was supportive of publication with some suggestions for improvements, whilst Reviewer 1 indicated the results were too speculative and requested the paper has more evidence the model fits with observations, plus other theoretical justifications. Following my own assessment, I agree fully with the comments by both, and also share the concerns of Reviewer 1. Further, based on the reply to Reviewer 1 comments I was not convinced how the manuscript would be improved to adequately address the concerns. Therefore, it is recommended that the manuscript be re-considered after major revisions.*

*For the re-submission, I would suggest the structure of the paper is reorganised and believe this may address some of the concerns.*

**Reply:** We thank the Editor for the constructive and insightful comments. As the editor suggested, and to address some of the concerns of reviewer#1 regarding evidence that supports the modeling results, we have modified the organization of the manuscript to include the laboratory scale experiments within the main text. We emphasize the limitations of these laboratory results conducted in a shallow basin for validation of the reservoir scale model. Hence, the primary insights gained from the experiments were the systematic variations of external forcing (wind, radiation and combination) and their impact on energy partitioning, surface temperature and evaporation suppression of covered surfaces. These results improved our understanding of the top boundary conditions in the reservoir scale model results. In addition, we have changed the structure of the revised manuscript by separating "Results" and "Discussion" and hope that the Editor finds these changes useful.

- *The Lake Mead validation example appears out of context with the underlying motivation of the paper. The text refers many times to shallow reservoirs, and Figure 1 sets this context clearly. Yet Lake Mead is a very deep reservoir where the surface mixed layer is impacted by complex processes (and I note the simple diffusivity model appears to have an error in the thermocline depth by about 10 m during summer stratification). I find it hard to reconcile that validating the*

*uncovered model on a reservoir of this size is also proof that the model is able to capture heat fluxes from a 3 m deep reservoir (covered or uncovered). Further, this un-covered model validation is presented as a result (Figure 4; Section 3.1), but this does not clearly align to an aim presented in the introduction, and seems better suited to an appendix - i.e. supporting material to help bench-mark the model prior to its application here. A second bench-mark simulation on a "shallow" reservoir would greatly help strengthen this argument - that your (uncovered) model is a reasonable tool for both shallow and deep reservoirs. Accessing public thermistor chain datasets through something like the GLEON network may provide a case-study?*

**Reply:** We thank the Editor for raising this point. Even if covered reservoir scale data were available we would have to perform a validation of the energy balance and temperature profile ingredients of the model as a reference state for comparison with the covered surface scenario. The use of the complete data set of Lake Mead provides a basis for this critical validation step (within the limits of a 1D model); we thus consider this element as an important ingredient of the study and not simply an appendix. The discussion of shallow reservoir behavior is an analysis of the model (without validation yet) for a range of boundary conditions to address the effects on heat storage and surface fluxes of reservoirs of different depths. Much is left to be studied about the effect of floating covers on water bodies and we slowly wade to this area using data from lab experiments in a shallow basin, and two ongoing field studies in deeper ponds under natural conditions (see images below). Even for the relatively well-studied behavior of uncovered water reservoirs, finding complete and definitive dataset for model testing was not a simple task (atmospheric forcing, temperature profile, surface fluxes, etc. …) and we felt fortunate to be able to use the Lake Mead data for model evaluation (even if many reservoirs of practical interest would likely to be shallower).

- *Following on from the comment above about the structure, the aims of the paper should more clearly align to the results and key discussion points. Currently, the connection is not always obvious - for example, there are two sections in the "Results and Discussions" that relate to ecology/gas fluxes (Section 3.3) and economics of covers (Section 3.4), yet neither of these sections present actual results, and also neither of these sections match the objectives listed in the introduction - usually sub-sections in the results/discussion should logically map to the objectives. Therefore, a revised submission should not use a combined results/discussion section, and better organise the sub-headings to separate results and discussion and to more obviously align with the objectives of the paper as they are argued in the introduction - i.e. every figure or sub-section should clearly align with one of the aims/objectives.*

**Reply:** Following this constructive comment, we changed the structure so that "Results" and "Discussion" are now separated and thus aligned the manuscript to suitably address main research questions of the work. In addition, we also added the "Materials and methods" section to the revised manuscript to better orient readers.

- *The Lab Experiment presented in Figure A1 is poorly integrated into the paper, currently just mentioned as a "motivation" for the model context and approach. In the reply to Reviewer 1 another Lab based figure is presented (this one has black discs whereas the first figure has white*

*discs). This is currently not clear - in the reply you state that you "incorporate insights" from your lab, and later state that the lab scale experiments "were in good agreement with model predictions for evaporation suppression and effects of cover color (as mentioned in page 15, line 3)" - however on Page 15 line 3 there is no specific comparison of the model with the lab data. I note it is stated it will be published a separate paper, however, that is not adequate for this paper. In the revision, the authors should better integrate the lab aspect of the study as a component of the paper (with a corresponding aim and result), or not mention it, or wait until the other paper is published. If it is included or referred to then the insights that link the lab and model should be clear rather than general statements that the agree. Related to my point above, I note that the validation of the Lake Mead is presented as a key result of this paper whereas the link with the lab results are an appendix - I would consider this should be the other way around given the aim of the paper is focused on covered systems.*

**Reply:** We have made changes in the structure of the revised manuscript by integrating the laboratory scale experiments into the main text and providing insights on processes at the covered surface of a reservoir (energy partition, evaporation suppression …) in support of model predictions. Considering the limitations of performing a quantitative comparison between "laboratory scale" measurements and "reservoir scale" modeling, we revised the manuscript to highlight the role of these different components of the study. The main role of the laboratory experiments is to provide direct evidence for cover efficiency and certain aspects of energy partitioning highlighting limitations to temperature profiles and associated feedbacks at the reservoir scale (i.e., heat storage, mixing and ground flux affecting water temperature and top boundary of the real reservoirs).

- *To address the lack of validation data for covered reservoirs, it may be worth refocusing the aims on exploring the potential sensitivities of the model to guide further experiments. For example, I found Appendix C to be an interesting result, yet it is only an appendix. More systematically demonstrating how the results vary in response to poorly known variables associated with the cover or lake will help make the model results more compelling, even in the absence of a full validation.*

**Reply:** We thank the Editor for this suggestion; hence such potential applications and merits of the modeling approach were highlighted in the revised manuscript.

- *The limitations of the model and needs for further research need to more comprehensively be discussed (in a discussion section). Currently there are aspects of this in the combined results/discussion. One issue raised by the reviewers, and is a big difference between Lake Mead and a small turbid reservoir, is the potential impact of non-neutral atmospheric boundary layer conditions - how would this impact the results? Others are raised in the discussion, and addressing them will enrich the paper*

**Reply:** We discussed limitations of the model in the "Discussion" section of the revised manuscript and provided suggestions for model improvements and further/future studies.

- *The summary needs to be re-written as a conclusion to more clearly align with the specific aims and highlight the most important findings and limitations/recommendations rather than repeat the study in general.*

  **Reply:** We thank the Editor for this important suggestion; the "Summary and conclusions" of the revised manuscript is now amended to better reflect major findings and limitations.

- *The paper needs to remove the speculative statements and more carefully word findings to match the specific results presented - for example, where it is stated the model was validated against the lab (page 15), or COMSOL simulations (page 13), this needs to be fully supported/justified rather than just asking us to trust and have faith that things are good. Statements such as "Floating elements efficiently suppress evaporative loss from water reservoirs and alter energy storage within the reservoir and oxygen exchange at the water-air interface" .... are not entirely supported by the current results and need to be much more carefully worded (there are no results related to oxygen exchange so how is this statement proven by this study?). To address the Reviewer 1 concerns, the paper must more clearly highlight that this study is assessing dynamics with a model that is yet to be validated on covered conditions.*

  **Reply:** Following the Editor's comment, we removed speculative statements not directly supported by our results in the revised manuscript and thus partly discussed them in the "Discussion". Moreover, we tried to better highlight our "primary" objective that was developing a physically based model for understanding energy dynamics and evaporation suppression of covered water reservoirs.

- *The paper needs a thorough review for grammar and expression, as many proof-reading errors occur throughout.*

  **Reply:** We have checked the manuscript thoroughly to remove errors and to improve the language.

In the following we address point-by-point concerns and add clarifications concerning issues raised by the reviewers.

Response to Reviewer # 1

- *First, at best, this investigation is speculative as there is no verification data presented to show whether or not is predictions are robust. At the simplest level, it would be anticipated that if a reservoir was covered in such a way as to prevent the penetration of electromagnetic radiation into its surface by elements of low thermal conductivity, the surface thermal forcing would be reduced. Therefore the outcomes shown in Figure 6 are not surprising. The authors have elected to use meteorological data that does not appear to have been gathered in the vicinity of a reservoir and apply it to a "hypothetical" reservoir. The key question is the degree to which the predictions are reliable and the authors have not addressed this question. As stated on page 11, the authors do have access to a model reservoir that is described in Appendix A. It is difficult to comprehend why they have not verified their model for a system where they were able to make measurements.*

**Reply:** We agree with the reviewer that certain aspects pertaining to covered "reservoir scale" predictions remain tentative and would require experimental confirmation (we acknowledged the lack of publically available data from covered reservoirs for model validation on page 12, line 19 of the original manuscript). To bridge the information gap, we have tested several key features of the model using data from uncovered reservoir in response to natural conditions (USGS Lake Mead data – see Figure 5 of the revised manuscript). This was done to establish a reference state for evaluating the potential role of floating covers on energy balance and heat storage of covered reservoirs. We also incorporated new insights from laboratory experiments using floating covers in a "small basin". These laboratory evaporation experiments used a 1.44 m$^2$ basin with 0.16 m depth (as depicted in Figure A below) and yielded valuable information regarding energy portioning and evaporation suppression under various forcing. The limitations are of course the limited depth and absence of vertical temperature profiles and mixing and behavior of covered reservoirs under natural atmospheric conditions. We consider the study as a step towards application of sound physical principles to the modeling of evaporation suppression, where model validation for uncovered reservoir using multi-season measured data has been performed and (limited) insights from laboratory experiments for covered and uncovered basins feed directly into the top boundary conditions for the model. We thus think that there are sufficient ingredients to make this investigation not so "speculative". We think that the study offers quantitative predictions for the effects of covers (most are intuitive except perhaps the "surprising" similarity of white and black covers in suppressing evaporation at equal efficiency despite different energy partitioning pathways).

[Figure]

Figure A: The water basin with 1.2×1.2 m$^2$ surface area and 0.16 m depth at the end of a wind tunnel built in our laboratory (STEP - ETH Zurich); uncovered basin (left) and covered with black discs (right). The experiments were conducted for a series of boundary conditions with black and white covers (not shown here).

- *Equation (4) is the conventional expression for stress transfer at an uncovered surface in the absence of wind wave growth. On page 11, line 13 we are told that u\*$_a$ has been determined from bluff body theory. There is no discussion of the merits of combining these characterizations when their underlying assumptions are clearly at odds.*

**Reply:** Our main motivation is to address aerodynamic interactions of airflow with elements floating on water surfaces (for which very little is known unlike many studies on interactions with

wavy or bluff body covered solid surfaces). The situation is even more complicated considering the simultaneous phase change where heat and mass are exchanged at the evaporating water surfaces thus potentially affecting aerodynamic interactions over these partially covered water surfaces.

More specifically, the eddy thermal diffusivity in the vertical temperature equation (Eq. 1) is based on a relatively simple and physically-based formulation of Henderson-Seller [1985] (Eq. 3) that expresses eddy diffusivity as a function of friction velocity at the water surface. Note that Eq. (4) emerges from equality of shear stresses at an interface. We invoked a well-established theory of drag partitioning over rough surfaces developed by Shao and Yang [2008] and Nepf [2012] that has been recently evaluated by Haghighi and Or [2015] to quantify the friction velocity ($u_a^*$) of air and define the friction velocity at the water surface ($u_s^*$) based on Eq. (4). Furthermore, we tested this representation by considering the boundary layer thickness (a function of $u_a^*$ [Haghighi and Or, 2015]) obtained from direct measurements of mass loss from our water basin covered with floating discs (not presented in this study). Details of aerodynamic interactions between airflows and floating cover elements are key to evaluate evaporation suppression and thermal effects in covered reservoirs and deserve specially designed studies (beyond the scope of the present work). Additional details on the friction velocity and boundary layer thickness in Appendix A thus aim to address such concerns.

- *In Figure 2, the authors invoke a conventional approach to the numerical modelling surface mixing of reservoirs which encapsulates unstable convection due to surface cooling. However, such an approach is unreliable in terms of heat fluxes and the authors' own observations with their infrared camera should show. Certainly the longstanding work by Andy Jessup and his collaborators have revealed very different behavior of the surface skin (responsible for radiant heat from the surface) from that of the bulk.*

**Reply:** We thank the reviewer for raising this important point. Equation (1) is a general energy balance equation for describing vertical temperature profiles developing in a water body or in a solid slab (with $D_w = 0$). What differentiates the solutions for these two scenarios are the eddy diffusivity and vertical mixing in water body triggered by thermal/density instabilities (e.g., a cold layer of water due to evaporative cooling overlying warmer water below). Such mixing processes are triggered at small scales diurnally (due to evaporative cooling at the surface), or seasonally where subsurface heat accumulation raises to the surface and unifies the vertical temperature profile in a reservoir (either Monomictic or Dimictic reservoirs). We note that the simple vertical mixing approach of Dake and Harleman [1969] is designed to maintain the energy balance of the reservoir.

We agree with the reviewer that surface heat fluxes would be affected by the mixing scenario imposed (and probably the surface skin temperature). To minimize this effect, we imposed vertical mixing considering the "mean daily" temperature profile providing the initial condition at the beginning of next day. This step reduces transfer of heat towards the bottom of the water body. Consequently, the "instantaneous" values of surface temperature and surface heat fluxes are obtained directly from the temperature equation with the proper surface boundary conditions represented in Eq. (8) or (12), hence unaffected by surface thermal mixing as depicted in Figure 2.

This can be seen, for example, by comparing winter surface temperature of uncovered reservoir in Figures 6 and 7b of the revised manuscript. The good agreement between model predictions of vertical temperature profile in Lake Mead and measurements (Figure 5 of the revised manuscript) further confirms our modeling approach based on Dake and Harleman [1969] without affecting the calculation of surface temperature and thus surface heat fluxes represented in Table 2.

We note that the considerations of surface skin temperature is not unambiguously resolved by this treatment, yet, since the model ultimately aims to solve the full energy balance for the floating cover itself (with own radiative and thermal properties) the sensitivity to the exact water skin temperature for radiative transfer in covered reservoirs would be less important.

Response to Reviewer # 2

- *This is a well written and presented article. It provides a relatively simple but surprisingly comprehensive theoretical and physical basis of evaporation suppression from simple, shallow reservoirs from which more detailed work can emerge. It does this by comparing models of an uncovered reservoir to ones covered by white and black circular discs. A 1-D, column approach was used. I wondered why triangular covers were not considered as they have the potential of having no gaps between them (or much smaller ones than a disc).*

**Reply:** We thank the reviewer for the efforts and for the many insightful comments. As mentioned in page 10, lines 10-16 of the original manuscript, the aerodynamic resistance for vapor flux from water gaps forming between cover elements is governed by the combined effects of gap size ($a_g$), boundary layer thickness ($\delta$) and the lateral spacing between neighboring gaps. For very small gaps formed by polygonal covers, gap sizes could become smaller than the boundary layer thickness (the ratio of $a_g / \delta$ smaller than 1). This case may yield evaporation enhancement disproportional to size of the gap according to Eq. (13) [Schlunder, 1988; Shahraeeni et al., 2012]. In addition, for certain applications of multiuse reservoirs, water gaps formed between spherical or cylindrical covers allow light penetration and surface oxygen transfer; both play important roles in ecological aspects of the water body.

- *The paper could be well served by articulating right at the outset the methodology you use. This is how I perceive it (from reading p. 11): 1. Calculation of evaporation reduction due to discs; 2. Effect of heat balance of the discs on water column, the primary evaporation reduction element; 3. Effect of heat balance of the gaps between discs on water column, including conduction from disc to water; 4. Effect of the increase of gap water surface temperature due to 2 and 3.*

**Reply:** We thank the reviewer for the suggestion. A summary of the main steps and methodology is provided in page 10, lines 5-9 of the revised manuscript.

- *Advection of (likely) colder water into the column was brought up in a discussion of managed input vs output for the reservoir but non-advective heat transfer was only considered for the bottom of the column. What about the four sides (can assume a simple soil temperature profile)?*

**Reply:** Clearly, for small reservoirs lateral heat exchange with the water body could be important in the energy balance. At this stage we seek to establish a simple 1D model for reservoirs where vertical temperature profile and surface heat fluxes dominate the response in the presence of floating elements, we thus neglect lateral heat transfers of the reservoir assuming that the side area of the reservoir is small relative to its surface area (as is likely in many shallow reservoirs). Following this comment, we explicitly mention this key simplifying assumption in page 8, lines 6-10 of the revised manuscript.

- *The diffusivity coefficient, D, did not appear to include any internal dynamics such as non-linear and/or breaking waves, which would likely increase it. The authors might consider such inclusion for completeness. Although, I must admit, internal motions in such a shallow reservoir would not be very large or complex. However, I am not aware of any observations of internal motions in shallow reservoirs and there are few for larger, deeper ones (with bottom topography forcing the wave motion). Managed releases would exacerbate wave activity.*

**Reply:** Some of the internal motions in deep reservoirs are attributed to the onset of thermal instabilities as included in the model representation (Eq. 10). Additionally, the effects of wind friction velocity are included explicitly in the (nonlinear) formulation of the eddy diffusivity in Eq. (3). Clearly, inflows-outflows, bottom topography, and breaking waves would enhance mixing and thus modify effective eddy diffusivity. However, keeping with the simple 1D formulation of Henderson-Seller [1985], we retain surface interactions of eddy diffusivity with wind (that is likely to be altered in the presence of the floating cover!). Following the reviewer's comment, we better highlighted these important aspects in the "Discussion" of the revised manuscript.

- *It appeared implicitly assumed that the water was not turbid, a rare condition in most reservoirs. A short discussion of the effect of turbidity on the columnar distribution of heat would enhance the work and provide an avenue for further theoretical work.*

**Reply:** We thank the reviewer for raising this point; we have considered parameterization of light penetration and various radiative effects in Eq. (2). However, we added a short discussion of this aspect in page 5, lines 3-5 of the revised manuscript.

- *While the amount of open water subject to heating is small in this study, for completeness at least a nod to the Clausius-Clapeyron relationship should be noted (and, I guess, dismissed). It had a major impact on the "failure" of monomolecular layer cover evaporation suppression in the famous Lake Hefner (Oklahoma, USA) Evaporation Reduction Experiment in 1967 (Bean and Florey, 1968, Water Resources Res., 4, 206- 208; also notes an evaporation reduction of about 60%) because the water warmed up when evaporation was reduced. Wind removed the layer, exposing the warm water, which then had higher evaporation due to the warmer water resulting in a net loss.*

**Reply:** We thank the reviewer for raising this important point. We include the Clausius-Clapeyron relationship in the representation of evaporative flux based on the "saturated vapor concentration" that is a function of surface temperature (e.g., see Eq. 8). The difference with the cases mentioned, is with the energy balance over the covers either via albedo reflection (white covers) or sensible

heat exchange (black covers) with minimal net heat flux to the surface (laboratory experiments). We added a discussion of the potential nonlinear evaporation enhancement effects as a function of surface water temperature in page 6, lines 20-22.

- *An important metric, the mean depth, D = V/A, where V is the reservoir volume and A, its surface area was not discussed. An efficient reservoir would be one where V is large and A is small resulting in a large value of D; in other words a cylinder will evaporate less than a bowl of the same volume. In this case 3m < D < 10m was considered. This is very shallow, implying a rapid response of reservoir heat content to varying atmospheric forcing; in other words the surface temperature, the main driver of the evaporative process, responds rapidly to latent and sensible heat transfer as well as the mean temperature of the volume. There is little phase lag between the near surface heat balance and interior heat balance; both will closely follow the daily average air temperature and net radiation input.*

  *In a deeper reservoir, Lake Mead was used where D is 165, there is a considerable phase lag in the diurnal and seasonal variations of surface versus interior temperature. For instance, in summer daytime air temperature will likely exceed the water temperature; a stable situation resulting in reduced evaporation especially in windless conditions. The reverse is true at night, when water temperatures are likely warmer than air temperature. Since during summer mid-latitude daylight hours substantially exceed nighttime hours so the lower evaporation during the day will dominate. In Fall, surface temperature will decrease due to lower insolation amount and duration, but will this will likely be mitigated by heat transfer into the surface layer by relatively warmer water in the interior resulting in relatively warmer surface temperature than air temperature throughout the day resulting in potentially more evaporation in that season (and Winter) compared to summer. The results shown in this article do not support this heuristic argument. However, eddy correlation observations over a period of years over Lake Superior (Blanken, P. et al., 2011, J. Great Lakes Res., 37, 707-716) show this nicely.*

**Reply:** The present model was developed with relatively shallow reservoirs used for seasonal storage in mind (i.e., depth<10 m). As pointed out by the reviewer, the response of such shallow reservoirs to atmospheric conditions is rapid and the time lag between surface and interior heat balances would be relatively short. A comparison with data from Lake Mead (a relatively deep reservoir), enabled testing of some key features of the model towards establishing a reference uncovered surface for evaluating effects of floating covers on the energy balance (the main objective of the present study). We agree with the reviewer that in the presence of phase lags and multiple mixing (e.g., dimictic reservoirs), the evaporative flux could be affected and even leading to enhanced evaporative losses during the winter. We point out however, that the monthly evaporation data from Lake Mead support the results of higher evaporative losses during summer even for such a deep reservoir (the Figure B below).

[Figure]

Figure B: Comparing model estimations of evaporative loss from Lake Mead with measurements demonstrating that evaporative losses during summer are dominant.

- *Last, I recall talking with a farmer who was the leader of a ditch company that managed a small reservoir as assumed here. He was very interested in estimating evaporation and, of course, suppressing it with some sort of cover as described here. I asked him if he had planted a wind break on the windward side. He was stunned and said he had not thought of it. So I said: "But you thought of it for your fields and that isn't open water. Furthermore, it would be a good use of otherwise "lost" leakage to ground water." So while I understand this windbreak approach and the consideration of internal boundary layers formed by changes in surface friction is not conducive to such a study as outlined here, I feel a theoretical approach to these aspects of real world reservoirs would be worthwhile in the search for low-impact geoengineering of simple reservoirs. This group obviously has the tools and expertise.*

**Reply:** We thank the reviewer for sharing these insights; we also consider aspects of internal boundary layer and its impact on local heat and mass transfer processes key to the efficiency of the cover. We thus plan to further investigate such nuanced aspects in the next steps of this ongoing project to provide a comprehensive framework, including both physical and ecological aspects for design and management of (optimal) floating elements.

**Minor comments**

- *p. 2, l. 15: I believe the recent use of black balls in a Los Angeles reservoir was not aimed at evaporation reduction but the reduction of toxic algae blooms. I think Israeli engineers have used white ping-pong like balls to reduce evaporation in test reservoirs (don't have a reference).*

**Reply:** The reviewer is right, the initial motivation was suppression of photochemical reactions and evaporation suppression from Los Angeles reservoir was a secondary goal. Nevertheless, the water saving aspect gained prominence with the lingering drought in California (as reflected in highly

publicized media cover: https://www.engadget.com/2016/09/21/the-big-picture-shade-balls-los-angeles-reservoir/).

- *p. 7, eq. 6a: Please check for references for some of these empirical relationships. Some equations are referenced, some not.*

  **Reply:** As mentioned in page 6, line 9 of the original manuscript, Eq. (3) and subsequent equations used for quantification of eddy thermal diffusivity are provided by Henderson-Seller [1985]. Following the comment of the reviewer, we explicitly pointed it in the revised manuscript.

- *p. 7, eq. 6b: some readers will not recognize the Brundt-Viasala relationship, which carries some restrictive assumptions with it. Interestingly on a windless or low wind day, this might be more likely during the day and convective mixing, as noted in this work, which is more likely at night when surface temperature might be lower than temperatures below.*

  **Reply:** We thank the reviewer for the comment; the Brunt-Vaisala relation is part of the stability parameterization of the eddy thermal diffusivity by Henderson-Seller [1985] and plays an important role in the water mixing we consider in this work.

- *p. 7, l. 12-13: Do you have a reference for the assumption?*

  **Reply:** The assumption arises from the continuity of temperature profile at the interface of liquid and solid phases. We provided appropriate reference in page 6, line 10 of the revised manuscript.

- *p. 8, eq. 8: explain why you use C for vapor concentration instead of the more recognizable q, specific humidity.*

  **Reply:** The representation based on the vapor concentration arises from Fickian mass transfer across the air boundary layer (implying dominance of diffusive fluxes [Haghighi et al., 2012]). In any case, we do not expect this to affect the clarity of the analysis as vapor concentration and specific humidity are linked via air density (we added a comment for the readers more comfortable with $q$ in page 7, line 1).

- *p. 8, l. 15, Fig. 2: is this the heavy dashed line in the Figure? It needs to be explained.*

  **Reply:** It is represented by the solid line denoted as $T_m$, we removed the heavy dashed line in the revised manuscript to avoid confusion of the readers.

- *p. 10, l. 13-14: jargon alert! "three-dimensional vapor shells" Show or explain further. Also what is meant by "lateral spacing"? Perhaps you can show these in Fig. 3b.*

  **Reply:** We thank the reviewer for alerting us of the use of such jargon. The point was that for small lateral spacing between neighboring water gaps in the cover (either for covers made up of small elements, or densely punctured plastic cover), the vapor concentration profile resulting over the surface is nearly 1D and layered (left image below); as spacing increases, the vapor profiles form individual 3D domes that act to enhance evaporative flux from individual gaps. This is schematically represented in the image below. We thus removed the expression in the revised manuscript to avoid confusion of readers.

[Figure]

**1D to 3D**

Figure C: Conceptual image of evolution of vapor shells above individual water gaps with increasing spacing between them.

- *p. 12, l. 12: Consider "Given the simplifying assumptions, the model overestimates: : :.".*

**Reply:** Thanks, we amended the sentence.

- *P. 13, Fig. 5: Comment on the slow uptake of heat in Spring (cold water/warm air) vs rapid decrease in Fall (warm water/cold air) to add confidence in the model. You might find observational evidence to back up a heuristic argument: surface layer more stable in Spring, more convective in Fall.*

**Reply:** We provided further discussions on the evolution of temperature profile in page 12, lines 18-23 of the revised manuscript.

- *p. 13, l. 18-19 "..demonstrate : : : a much colder reservoir." This is an impressive modeling result and should be tested by a field experiment. Is one being considered?*

**Reply:** We are aware that model predictions require confirmation using reservoir scale experiments that are currently unavailable. We note that a colder water body under floating elements was observed in our preliminary lab scale measurements in a small basin ($1.2 \times 1.2 \times 0.16$ m$^3$), and are presently conducting two field scale measurements (in EAWAG (Switzerland) and Isfahan University of Technology (Iran); see images below) to provide the necessary data for model evaluation.

[Figure]

Figure D: Top: ongoing "field scale" experiments conducted in EAWAG near Zurich using 8 ponds each 14 m$^2$ and 1.5 m deep covered with white and black 0.2 m (EVA foam) floating

covers including two uncovered control ponds; bottom: construction of two reservoirs (5×5×2 m$^3$) in Isfahan for field scale tests of the model in a dry and hot place with significant evaporative demand.

- *p. 17, Section 3.3, Ecological considerations: Reservoirs, even small, simple ones as assumed here, while not likely used for recreation, can be important to migratory birds and other wildlife as well as aquatic life in the reservoir (which often provide food for wildlife visiting the reservoir, extending the ecological boundary). Discs, as described here, will inhibit access for wildlife. That should at least be mentioned along with the impossibility of modeling it. Although, for any future work, you might consider entraining a wildlife expert who might.*

  **Reply:** We agree with the reviewer that ecological considerations of covered reservoirs are not limited to aquatic organisms only and additional biological agents such as birds and wildlife should be considered. Future development of the framework will consider other more nuanced ecological aspects such as optimizing surface coverage to provide required light and oxygen for aquatic life and accessibility to other organisms as pointed out. We thus mentioned this important point in page 18, lines 18-20.

- *p. 18, Section 3.4, Costs and water savings: A nice summary. Have an economist vet it, if you haven't. I especially liked the last sentence. You might mention water scarcity as a conflict enhancer as described recently by Tom Friedman, a well-known columnist for the New York Times. So efficient storage of water becomes political.*

  **Reply:** We thank the reviewer for raising this point and further discuss (the rapidly evolving) economics of water saving in the revised manuscript and in follow up studies.

- *p. 22, l. 10-18: You should mention an important effect of the discs I did not see in the paper but suggested in this discussion; the appearance of waves, breaking waves, and spray as wind increases (threshold _ 6 mps). This radically changes the situation in open, uncovered water and greatly increases the evaporation; modeling this effect is still elusive though a check of hurricane boundary layer modeling may provide some insights.*

  **Reply:** We discussed the impact of discs on surface waves and shear velocity and potential effects on the evaporative loss in the "Discussion" of the revised manuscript. However, note that such aspects deserve comprehensive studies that are beyond the objectives of this work.

- *P. 23, Eq. B3: Is this correct? Should it be $\lambda = D/H$ to be dimensionless as described later in the Appendix?*

  **Reply:** Please note that the dimension of $d.H$ is m$^2$ and the parameter $N$ represents number of discs per unit "area" rendering $\lambda$ a dimensionless parameter ($\lambda = N\,d\,H$).

- *p. 24, l. 13-15: 3 to 10 m is not enough depth variation. Note that Rn, H, and E are essentially constant and heat storage decreased by 27% for the 10 m depth. Can you show the "index" Ta – Tw for the two depths? I predict they will be nearly the same.*

**Reply:** Noting that the focus of the study is ultimately on evaporation suppression and behavior of shallow reservoirs often used for seasonal water storage in arid regions. The similarity in surface heat fluxes between "shallow" and "deep" scenarios considered here is associated with similar surface temperatures and the effect of depth is thus reflected in specific storage and bottom fluxes. Following the reviewers comment, the plot below depicts the difference between air temperature and mean vertical temperature of the reservoir. Although they look similar, the differences in summer and winter are of the order of 3°C.

[Figure]

Figure E: Variation of $T_a$-$T_w$ for shallow and deep reservoirs. The water temperature ($T_w$) represents the mean vertical temperature of each day.

- *Figure 1: I'd replace that with a Google Earth picture of the Front Range (eastward) of Colorado which is dotted with small reservoirs to show how ubiquitous they are. Using the area tool on the USGS National Map Viewer you could show that the combined area of these "small" reservoirs approximate that of major reservoirs in the Colorado River Basin system.*

**Reply:** We thank the reviewer for the suggestion; we are presently conducting a global survey of the characteristic of such on-farm reservoirs in different regions (completed surveying NE India, Italy and Australia). The survey would place the use of such storage in the context of other free water storage and highlight the potential evaporative losses (and potential usefulness of floating covers to suppress these losses). We opted to keep the images for simplicity (not much different than the images from the Front Range…).

- *Figure 2: Those two hatched areas do not look equal to me. Explain the dashed line.*

**Reply:** Considering Eq. (10), the hatched areas on the left and right hand sides of $T_m$ are the same. As mentioned earlier, we removed the dashed line in the revised manuscript to avoid confusion of potential readers.

- *Figure 3: What is the red triangle on the far left side? You've labeled the down arrows to the far left and right, what is the label for the one between them? What does the expression below fc represent?*

**Reply:** The red triangle stands for the free water surface that is a standard sign in the literature. The arrow between the down arrows is $q_c$; we added it to the revised figure. The expression indicates attenuation of radiative flux in depth; we better explained it in the revised manuscript.

- *Figure 4: "assumed"? Be honest, wasn't it "tuned"? Were "η" and "β" observed?*

**Reply:** To obtain the radiative properties and light attenuation coefficient in Lake Mead, we contacted Dr. Michael Moreo at USGS who is responsible for measurements at the lake; his reply was "*I do not have any subsurface radiation attenuation data. I will say the lake is very clear, and I suspect that radiation penetration is as deep as other very clear lakes*". We thereby estimated the values of $\eta$ and $\beta$ based on the literature data reported for clear water bodies.

- *Figure 7: I think 7c is a result of the shallowness of the reservoirs you are modeling. They are like an evaporation pan which has a similar trace with respect to season. Deeper reservoirs show a maximum in Fall/Winter and a minimum in Summer for good reasons. Dew forms on the surface of Lake Superior in summer!! I've witnessed explosive evaporation events associated with reservoir overturning in mid-winter with air temperature of -12C.*

**Reply:** Please note that higher evaporative loss during winter is not general as observation in Lake Mead indicates higher evaporation during summer (please see Figure B above). Considering the reviewer's comment, we thus recall that the focus of the present work is on relatively shallow reservoirs whereby evolution of surface fluxes are expected to follow seasonal cycles with higher evaporative loss in summer.

We thank again the reviewers for many helpful comments and hope the Editor finds the clarifications satisfactory.

Sincerely,
Milad Aminzadeh, Peter Lehmann, and Dani Or

[revised manuscript text omitted]

---

## Editor Decision (ED1)

[revised manuscript text omitted]

*this first sentence is for the introduction, not the methods*

*not clear what exactly FLUXNET is - is it meteorological data? add details*

As mentioned above, not much is known on evaporation suppression and energy balance of partially covered water bodies in reservoir scales. We thus opted for synthetic test of the model using FLUXNET data from Majadas (Spain), a dry region with significant atmospheric evaporative demand for the water year from March 1, 2004 to March 1, 2005. We then employed the model and considered potential effects of floating disc-shaped elements (diameter of 0.2 m and thickness of 0.02 m) on heat fluxes and water temperature profiles within a hypothetical reservoir with a depth of 10 m using half-hourly meteorological data.

*how was the model setup? grid? parameters?*

**4 Results**

**4.1 Energy budget and evaporation from uncovered water reservoirs – model application**

The theoretical model for quantification of water temperature profile within the water body and surface

*this sentence is methodological; it is repeating what was already stated in the methods.*

fluxes was evaluated using vertical water temperature and surface energy balance components obtained from measurements at Lake Mead (Moreo and Swancar, 2013). Figure 5 compares our model

*Try not to open a sentence with a Figure number; instead it is better to make a statement of finding and then refer to the Figure as support.*

estimations of mean monthly temperature profiles with water temperature measured at different depths (Moreo and Swancar, 2013). The assumed thermal and radiative properties of the lake (and covers) are

listed in Table 1. The measured and simulated fluxes are summarized in Table 2 showing good

agreement between modeled and measured annual net radiation ($R_n$) and evaporation ($LE$) fluxes.

5    Given the simplifying assumptions, the model overestimates the reported sensible heat flux ($H$) that

was estimated by Moreo and Swancar (2013) based on the Bowen ratio method with associated energy

closure considerations (Foken, 2008; Kalma et al., 2008).

These results for temperature profile and energy fluxes of uncovered water reservoir primarily provide a

"reference state" for investigating effects of partial coverage on heat storage and energy balance of

10   reservoirs.

**4.2 Laboratory evaporation suppression experiments**

We conducted laboratory experiments focusing on the surface-cover-air interface for different forcing to

capture energy partitioning dynamics under steady state conditions and mimicked diurnal cycles and

thus provide a basis for the modeling assumptions. Surprisingly, the measurements depicted in Figure

15   6a have shown that irrespective of the type of external forcing (wind and radiation) or the color of the

floating discs, the resulting evaporation rate from covered water surfaces was about 20% of the

uncovered surface. The corresponding evaporation suppression efficiency of 80% is less than the cover

fraction of 91%. This reduced efficiency is attributed partially to the increased surface temperature of

the water between the discs compared to uncovered water reservoir (shown in Figure 7). Subsequently

20   we have used the laboratory forcing (wind, radiation, air temperature and humidity) in the model to

describe the evaporative losses and capture temperature dynamics over the surface of uncovered and

covered water basin. Despite the small size of the basin (and scale mismatch with reservoir scale

model), the simulations were in good agreement with mass loss rates measured by the balance (Figure

6b), and surface temperatures recorded by IR thermography (Figure 7). This limited experimental

25   evidence highlighted the potential applicability of the model for quantifying evaporation suppression

and predicting dynamics of surface temperature that is in the core of energy partitioning over covered water bodies.

The theoretical results supported by laboratory measurements clearly demonstrated that the main effect of floating covers on evaporation suppression and energy partitioning was concentrated at the surface and thus upheld the focus on the top boundary condition for the full reservoir scale model reported in this study. Nevertheless, we note that these experiments may not reflect influences of temperature profiles, heat storage, mixing and ground flux that could potentially affect water temperature and, in turn, the top boundary of the reservoir. **this is more of a discussion paragraph, rather than results per se**

**4.3 Evaporation and energy budget of partially covered water reservoirs**

Covering a water reservoir with floating elements affects absorption of radiative energy at the surface and consequently the water body heat storage and temperature profile. Considering the lack of field scale data from partially covered water reservoirs, we evaluated the model for a hypothetical reservoir using meteorological data obtained in Majadas, Spain. **this paragraph has content that should be covered in the introduction and methods, not the results**

[revised manuscript text omitted]

---

## Author Response (AR2)

**Response to Reviewers' Comments (manuscript # hess-2017-415):**

**"Evaporation suppression and energy balance of water reservoirs covered with self-assembling floating elements"**

**Milad Aminzadeh, Peter Lehmann, and Dani Or**

Dear Dr. Hipsey,

We greatly appreciate the many insightful comments made by you and the reviewers. Following your valuable suggestions, we have made a comparison between the proposed model and our own laboratory measurements conducted in a small basin (despite the large disparity in scales as explained next).

Previously, we have been reluctant to apply the reservoir scale model to our laboratory scale measurements not only due to the scale mismatch but also potential issues with the parametrization of the governing equations that consider mixing in large reservoirs. We were pleasantly surprised that model predictions were reasonably consistent with lab-scale observations for covered and uncovered basins (as depicted in Figures 6 and 7 of the revised manuscript). We have taken a closer look at the value of eddy thermal diffusivity as the primary candidate for potential discrepancy between shallow basin (0.16 m depth) and natural water bodies (depths of many meters) in terms of their mixing dynamics thermal characteristics. The attached Figure 1 depicts effects of reservoir depth on the effective eddy thermal diffusivity (Eq. 3 in the manuscript). The shallow basin restricts formation of larger eddies and thus yields lower values of eddy thermal diffusivity (often in the order of $10^{-4}$ $m^2$/s in natural water bodies such as lakes). Hence, wind-driven mixing in our basin resulted in an eddy thermal diffusivity of the order of $10^{-5}$ $m^2$/s, an order of magnitude smaller than natural water bodies (recall that the molecular thermal diffusivity of water is $1.43\times10^{-7}$ $m^2$/s,).

We have made additional revisions (marked in blue) to incorporate these new changes in the manuscript (in addition to the new comparisons and laboratory results). We sincerely appreciate the constructive comments made by you and reviewers since last summer and strongly hope you will find these new changes in the manuscript and clarifications satisfactory.

Sincerely,
Milad Aminzadeh, Peter Lehmann, and Dani Or

[Figure]

**Figure 1. The effect of reservoir depth on variation of eddy thermal diffusivity for the same forcing (wind speed ~ 2.3 m/s).**

[revised manuscript text omitted]

---

## Author Response (AR3)

**Response to Reviewers' Comments (manuscript # hess-2017-415):**

**"Evaporation suppression and energy balance of water reservoirs covered with self-assembling floating elements"**

**Milad Aminzadeh, Peter Lehmann, and Dani Or**

Dear Dr. Hipsey,

We greatly appreciate your insightful comments. Following your valuable suggestions in the pdf file, we have improved the quality of the text and made some changes in the "Materials and methods", "Results" and "Summary and conclusions" to address the highlighted points in the review file. We have expressed more details and added justifications in the "Materials and methods" to better clarify different aspects of the modeling setup for potential readers of the manuscript.

We sincerely appreciate all your constructive comments and hope you will find these new changes in the manuscript and clarifications satisfactory.

Sincerely,

Milad Aminzadeh, Peter Lehmann, and Dani Or

[revised manuscript text omitted]